# Phytophagy impacts the quality and quantity of plant carbon resources acquired by mutualistic arbuscular mycorrhizal fungi

C. A. Bell ●[1] ✉, E. Magkourilou[1,2], J. R. Ault ●[3], P. E. Urwin ●[1] & K. J. Field ●[2]

Arbuscular mycorrhizal (AM) fungi associate with the roots of many plant species, enhancing their hosts access to soil nutrients whilst obtaining their carbon supply directly as photosynthates. AM fungi often face competition for plant carbon from other organisms. The mechanisms by which plants prioritise carbon allocation to mutualistic AM fungi over parasitic symbionts remain poorly understood. Here, we show that host potato plants (*Solanum tuberosum* cv. Désirée) selectively allocate carbon resources to tissues interacting with AM fungi rather than those interacting with phytophagous parasites (the nematode *Globodera pallida*). We found that plants reduce the supply of hexoses but maintain the flow of plant-derived fatty acids to AM fungi when concurrently interacting with parasites. Transcriptomic analysis suggest that plants prioritise carbon transfer to AM fungi by maintaining expression of fatty acid biosynthesis and transportation pathways, whilst decreasing the expression of mycorrhizal-induced hexose transporters. We also report similar findings from a different plant host species (*Medicago truncatula*) and phytophagous pest (the aphid *Myzus persicae*). These findings suggest a general mechanism of plant-driven resource allocation in scenarios involving multiple symbionts.

Arbuscular mycorrhizal (AM) fungi form symbioses with the roots of most plants across nearly every ecosystem on Earth[1]. These partnerships usually impart multiple benefits to the host plant, including increased growth and disease resistance[2,3], often as a result of AM fungi improving host plant access to and acquisition of soil nutrients[4]. AM fungal-acquired nutrients, including nitrogen and phosphorus, are transferred to the host plant, while AM fungi acquire photosynthetically derived carbon-based compounds, including sugars and lipids[5–7]. As obligate biotrophs, AM fungi are entirely dependent on their host for carbon, lacking the ability to produce the necessary enzymes to degrade organic matter[8].

The mechanisms underpinning AM fungal acquisition of plant carbon are complex. AM fungal colonisation of plant roots induces vast transcriptional changes in the root tissues, including the induction of host genes that deliver carbon to AM fungi in the form of sugars[9]. Monosaccharide transporters (MST) and Sugars Will Eventually be Exported Transporter (SWEET) genes are important in the accumulation of sugars into cells containing AM fungal arbuscules[9]; a process which is potentially driven by an increasing sink strength imposed by AM fungal partners[10]. To reach their ultimate fungal destination, plant sugars pass through host plant membrane transporters into the peri-arbuscular space where they move through fungal MSTs that mediate the influx of sugars into AM fungal mycelium[11,12]. Silencing of a fungal MST gene impairs mycorrhizal formation[5], demonstrating the critical role of sugar transport in the establishment and function of the symbiosis.

AM fungal genome analysis shows that the genes encoding the molecular complex for fatty acid (FA) synthesis are not present,

[1]School of Biology, Faculty of Biological Sciences, University of Leeds, Leeds LS2 9JT, United Kingdom. [2]Plants, Photosynthesis and Soil, School of Biosciences, University of Sheffield, Sheffield S10 2TN, United Kingdom. [3]School of Molecular and Cellular Biology, Faculty of Biological Sciences, University of Leeds, Leeds LS2 9JT, United Kingdom. ✉e-mail: c.a.bell@leeds.ac.uk

suggesting they do not synthesise FAs, such as palmitic acid (16:0), de novo[13]. Instead, they are fully reliant on their hosts for their supply of FAs[8,14]. The occurrence of host C16:0 palmitic acid within AM fungal biomass and mirrored lipid profile between host roots and AM fungal partners[15] provide strong evidence that FAs make up a critical component of plant-to-fungus carbon supply, with AM fungi being only able to modify, rather than synthesise, these carbon-based molecules[16]. Upregulation of the host plant C16:0 FA synthesis pathway upon cellular differentiation into arbuscocytes (cells containing AM fungal arbuscules) appears to be driven by AM fungi-inducible promoter elements[6]. Impaired lipid transfer and arbuscule formation occur upon mutations in several plant FA synthesis genes[6,17,18].

Plants typically simultaneously interact with myriad other organisms that may compete with AM fungi for host plant resources - a scenario as ubiquitous in nature as plant-AM associations themselves. Such interactions impact carbon-for-nutrient exchange between plants and their AM fungal partners, for example, when a plant is exposed to foliar aphids[19,20] or plant-parasitic nematodes (PPN)[21], AM fungal symbionts receive dramatically less plant-fixed carbon than plants that are not exposed to the parasites. In both scenarios, delivery of fungal-acquired P and N to host roots is maintained resulting in the AM fungi appearing to contribute more to the partnership than they receive.

The persistence of AM fungi in systems where they may have limited access to plant carbon is intriguing, particularly considering their obligate biotrophic status. When supply of plant carbon is limited, AM fungi are compelled to establish symbioses with relatively poor, pathogen-infected host plants that have limited capacity to support AM symbionts. The amount of carbon transferred to certain AM fungi measured in these scenarios may represent the minimum amount required to maintain AM associations[21]. However, in natural environments, AM fungi would have the opportunity to form partnerships with other, possibly more 'generous', host plants potentially mitigating the limitations in carbon supply[21,22]. While there is evidence suggesting that neighbouring plants may at least partially compensate for disruptions in plant carbon supply to AM fungi caused by herbivory, they do not seem to completely make up for the shortfall caused by plant interactions with competing organisms[23].

Plants may also interact with additional symbionts that may be antagonistic, such as plant-parasitic nematodes (PPN) and foliar aphids that concurrently establish intimate and long-term interactions with their host. Both PPN and aphid phytophagy leads to a reduction in plant carbon allocation to AM fungi[19,21]. There are two potential explanations for this phenomenon, with important knowledge gaps in our understanding. Firstly, plants may actively withhold carbon from tissues that experience losses to symbionts, including AM-colonised roots, albeit with varying success depending on the symbiont identity[21]. Carbon allocation may then be optimised towards plant growth and defence mechanisms, and this is likely dynamic throughout the plants' lifetime. Alternatively, plant carbon could be redirected to partners with the greatest sink strength, which may well not be AM fungi, but rather the competing antagonistic symbiont[21]. The regulatory mechanisms governing the allocation of plant carbon to partnering organisms in either scenario are unknown, representing a significant knowledge gap in our understanding of how plants navigate and survive simultaneous competing resource demands from diverse partners.

To investigate the dynamics of resource allocation between plant roots and AM fungi during competing, antagonistic symbiosis (parasitic nematodes or aphids), we conducted split-root microcosm-based experiments where AM fungi (*Rhizophagus irregularis*) and plant-parasitic nematodes (*Globodera pallida*) colonised separate portions of the same potato (*Solanum tuberosum* cv. Désirée) root systems. This allowed us to separate and analyse the individual influences of competing symbiotic interactions on AM acquisition of plant carbon. Using a combined approach of $^{14}$C isotope tracing and metabolomics, we determined both the quantity and quality of plant carbon compounds allocated to AM fungal partners in the presence of PPN on host plant roots, respectively. To unpick the underlying regulatory mechanisms involved in plant resource allocation to AMF during symbiosis with different symbionts, we used transcriptomics to identify the key genes involved in mediating resource allocation to AM fungi and their responsiveness to concurrent competing interactions with PPN. To test the ubiquity of the regulatory mechanisms uncovered, we repeated the experiments using foliar aphids (*Myzus persicae*) as the competing parasitic symbiont on potatoes as well as *Medicago truncatula*, and determined the quantity and quality of plant carbon transferred to AM fungi. By integrating experimental approaches with transcriptomic analysis and considering the influence of host plant and symbiont identities, our study provides evidence for a dynamic flow of fatty acid and sugars to AM fungi from hosts exposed to parasitism from different pests. Our findings contribute understanding of the regulatory mechanisms underpinning resource allocation between plants and their myriad competing symbionts.

## Results

### AM fungal colonisation boosts plant growth, independent of PPN infection

Colonisation of plant roots by AM fungi had a consistently positive effect on shoot biomass, regardless of the presence or proximity of PPN to AM fungal-colonised root systems (Fig. 1A). Colonisation of host roots by AM fungi did not directly enhance photosynthetic efficiency ($F_V/F_M$) compared to asymbiotic control roots, however it did alleviate PPN-induced reduction in $F_V/F_M$ but only when colonising locally to the nematode (Fig. 1B). AM fungal colonisation increased root biomass compared to the asymbiotic control, whereas the biomass of PPN-only infected roots was reduced (Fig. 1C). This trend was observed in AMF-only and PPN-only inoculated roots of the same host's root system (Fig. 1C).

Roots colonised by AM fungi supported a greater number of PPN cysts per g soil (i.e. nematodes that invaded and developed through to cysts), as well as the number of eggs per cyst (i.e. number of eggs produced by each female cyst) (Fig. S1). This effect was most pronounced when AM fungi and PPN were inoculated within the same root compartment and interacting with nearby root tissues.

There was no difference in AM fungal colonisation of plant roots or hyphal growth into the soil whether PPN were present in the plant root system or not, regardless of whether PPN and AM fungi were inoculated into the same root compartment or were separated (Fig. S2). AM fungal-colonised host plants were richer in shoot P than non-AM fungal-colonised hosts, regardless of presence of PPN. Additionally, inoculating AM fungi into both sides of the PPN + AMF split-root compartments did not elevate shoot P more than hosts with AM fungi in only a single root compartment (Fig. S3).

### Carbon allocation to AM fungi is influenced by presence of phytophagous pests

More host-fixed carbon was allocated to roots colonised by either AM fungi or PPN, or both simultaneously, compared to asymbiotic plant roots (Fig. 2A). When PPN and AM fungi were inoculated into different root compartments of the same host ("PPN/AMF"), greater amounts of plant carbon were allocated to AM fungal-colonised root tissue compared to PPN-infected root tissue (Fig. 2A; B).

The amount of plant carbon lost to PPN when both AM fungi and PPN were inoculated into the same root compartment was approximately 2-fold less than when PPN were the only symbionts present (Fig. 2C). The amount of C acquired by PPN was further decreased when both PPN and AM fungi were inoculated into different root compartments (Fig. 2C). Similarly, the amount of plant carbon obtained by AM fungi was reduced by around 15-fold when PPN are

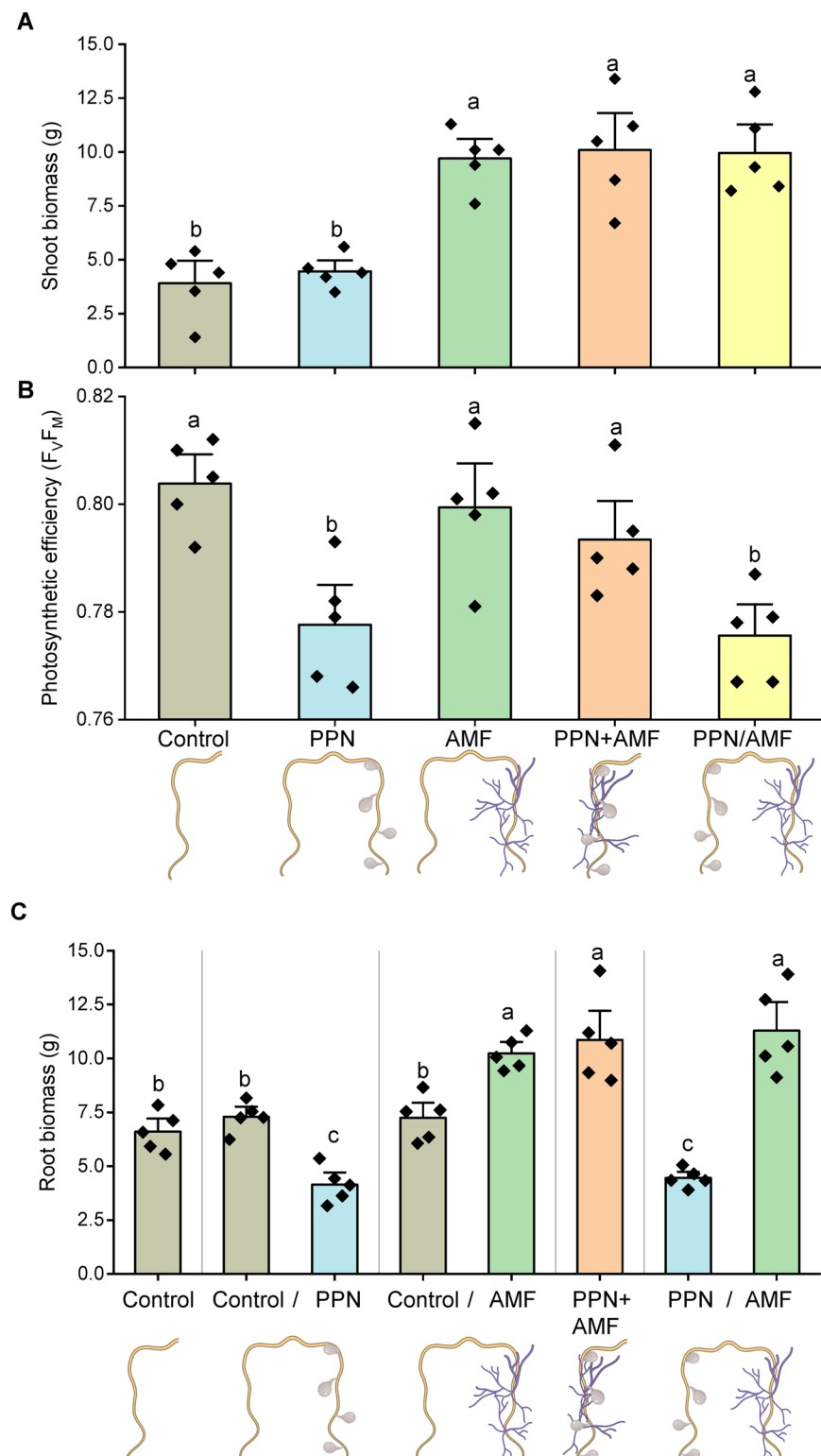

**Fig. 1 | The impact of AM fungi (AMF) and plant-parasitic nematodes (PPN) interactions on the plant host.** Data indicate the impact of AMF and PPN on mean **A** shoot biomass, **B** host photosynthetic efficiency before harvest as indicated by $F_V/F_M$, and **C** root biomass. Treatments are indicated by $x$ axis labels and schematics: control = no symbiont, '/' = roots split between the two stated treatments, '+' = both symbionts together on the same sampled roots. All data were collected at 5 weeks post inoculation. Root biomass indicates the biomass of roots in the specified split-root compartment. Bars represent five biological replicates with the standard error of the mean. Different letters denote significance (**A**, **B** one-way ANOVA, Tukey's Honest Significant Difference test, $p < 0.05$. **C** Linear mixed-effects models were applied to account for the non-independency of split roots belonging to the same plant, Tukey's Honest Significant Difference test, $p < 0.05$).

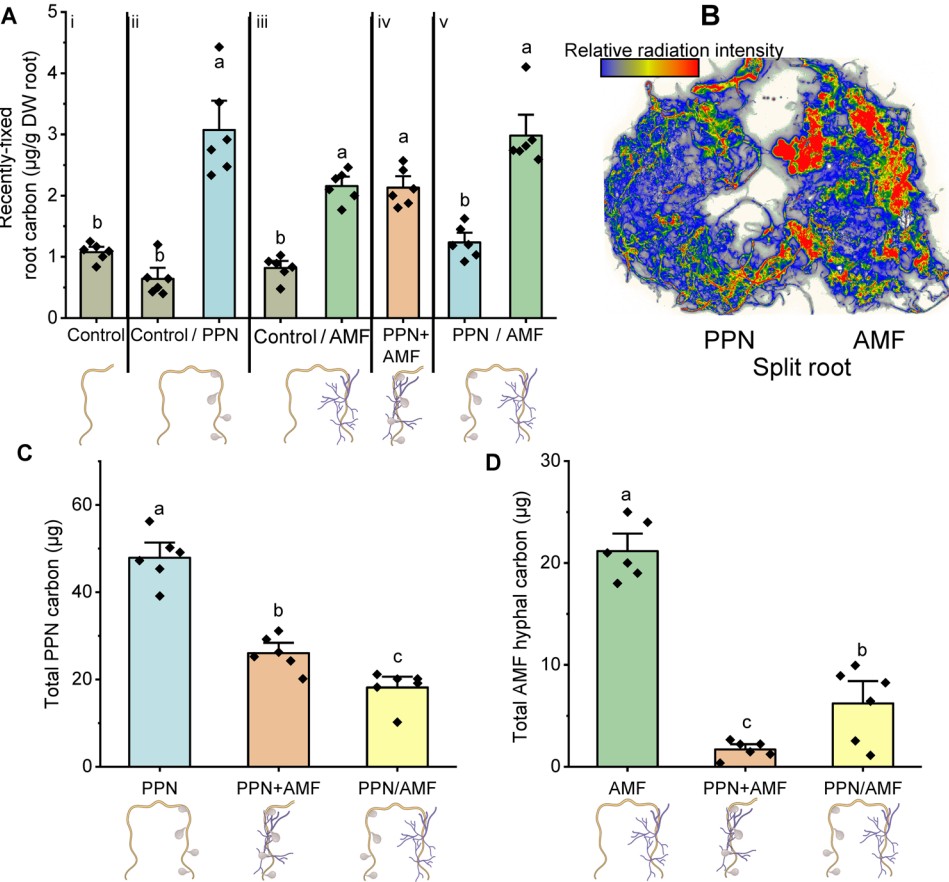

**Fig. 2 | The distribution of plant C following inoculation with plant-parasitic nematodes (PPN) and AM fungal (AMF) symbionts.** Mean amount and distribution of C within (**A**, **B**) roots, (**C**) PPN, and (**D**) AMF within each treatment. i–v in (**A**) indicate the separate split-root treatments, with data normalised per g of DW roots. **B** $^{14}$C distribution across a split-root system inoculated with PPN and AMF, where intensity of colour indicates relative accumulation of plant-fixed $^{14}$C. **C**, **D** quantify the amount of host carbon within the entire PPN populations and/or AMF hyphal mycelium within the pots. All data were collected at five weeks post inoculation.

Treatments are indicated by *x* axis labels: control no symbiont, '/' roots split between the two stated treatments, '+' = both symbionts together on the same sampled roots. Bars represent six biological replicates with the standard error of the mean. Different letters denote significance (**A** Linear mixed-effects models were applied to account for the non-independency of split roots belonging to the same plant, Tukey's Honest Significant Difference test, $p < 0.01$. **C**, **D** one-way ANOVA, Tukey's Honest Significant Difference test, $p < 0.05$).

inoculated locally (Fig. 2D). However, when both symbionts were inoculated into different root compartments, an intermediate amount of plant carbon was obtained by AM fungi (Fig. 2D).

### Differential gene expression controls allocation of plant carbon to AM fungi

AM fungal-colonised root tissue expressed a distinct gene expression profile compared to asymbiotic roots of the same plant (Fig. S4). Expression profiles of AM fungal-colonised root tissue in plants where PPN were inoculated into the distal root compartment show a greater similarity to asymbiotic control roots than AM fungi-only roots (Fig. S4).

Differentially expressed genes between these treatments were filtered to yield genes that have been previously identified as key for AM fungi-host resource exchange or represent potential candidates[5,6,8,9,11,17,24–26]. The expression of 4 sugar transporters, 8 *SWEETs* and 3 sucrose-related genes were upregulated in AM fungal-colonised roots compared to asymbiotic roots of the same host (Fig. 3, Fig. S5). The mycorrhizal-induction of these genes were reduced if hosts were distally infected by PPN (Fig. 3, Fig. S5). Overall, the mycorrhizal-induced upregulation of carbohydrate transport-related genes was reduced if PPN were infecting the opposing half of the root system.

We also quantified the expression of genes related to lipid biosynthesis and transport, and involved in host-AM fungi lipid transfer

pathways[6,7]. AM colonisation of roots induced the expression of 5 out of 12 genes that have been identified as key in host biosynthesis of lipids that may be exchanged with AM partners[6] (Fig. S6). Additionally, the two genes proposed to mediate transfer of synthesised lipids to AMF (*STR1/STR2*)[24] were also upregulated in roots colonised by AM fungi. However, in contrast to carbohydrate transporters, all of these remained similarly upregulated when PPN were inoculated into the second root compartment of the same hosts' root system (Fig. S6).

The expression of several host genes related to the transport of phosphate and nitrate, and potentially key for the intake of AM nutrients (e.g. *PT4*[27]), were upregulated in AM fungal-colonised roots compared to controls (Fig. S7). The expression of these genes was stable in AM fungal-colonised roots which had PPN-infecting partitioned roots from the same host, apart from *StPT4* which was reduced.

### Transfer of host-hexoses to AM fungal partners is independent of fatty acid transfer

Due to the differential expression of hexose-related, but not fatty acid biosynthesis/transport, genes, we quantified the amounts of C16:0 palmitic acid, glucose and fructose in AM-colonised roots and AM fungal hyphae either in the presence or absence of PPN (Fig. 4, Fig. S8). Palmitic acid concentration within AM-colonised or PPN-infected roots was greater than in asymbiotic roots (Fig. 4A, Fig. S8). The concentration of palmitic acid was greater in AM-colonised over PPN-

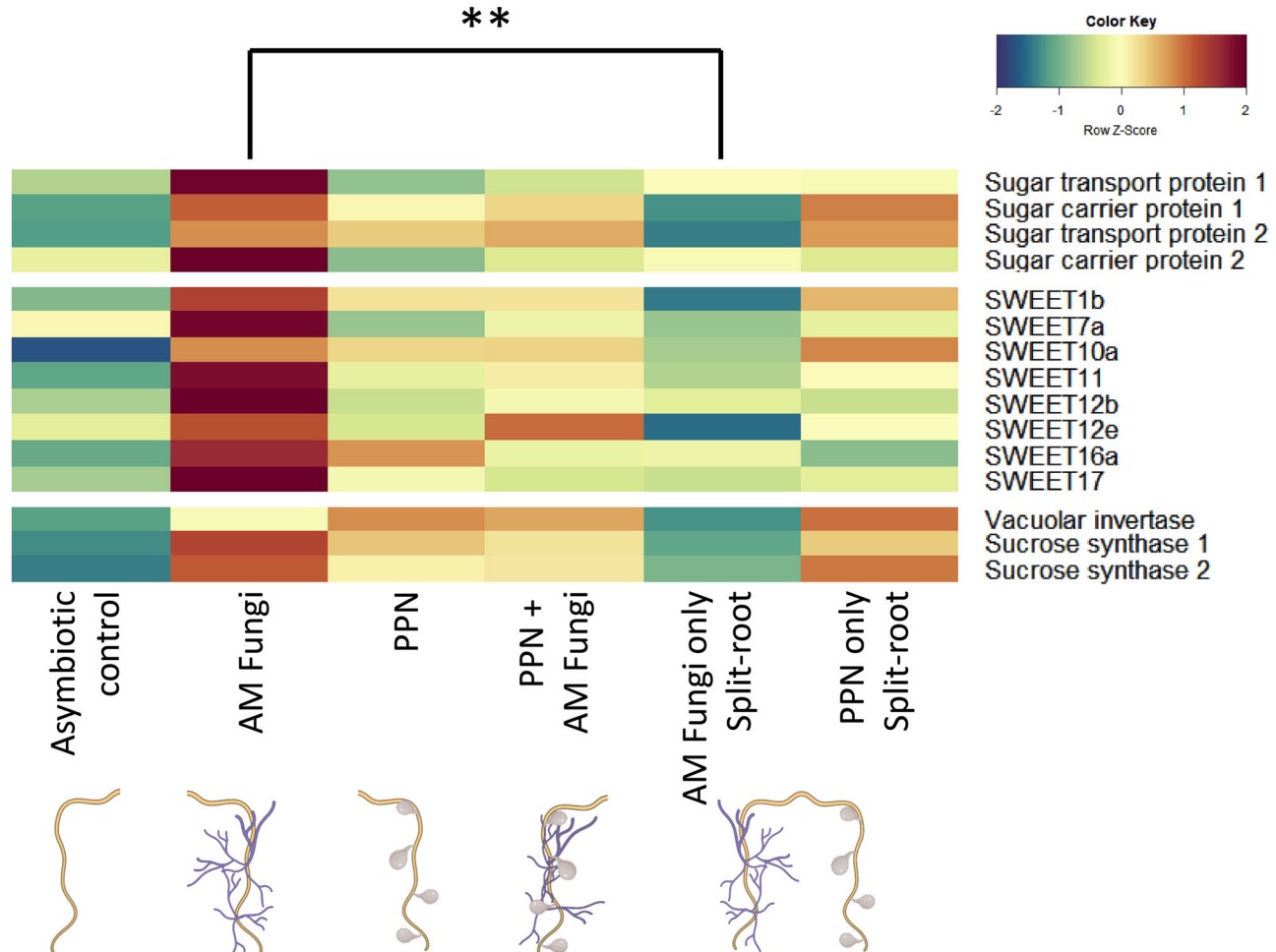

**Fig. 3 | Upregulation of genes related to the transport of carbohydrates in AM-colonised roots in the absence of PPN infection of the root system.** Genes are separated according to their role as/in monosaccharide transport, *SWEETs* (sugar transporters) or sucrose-cleavage. Heatmap was generated in R using heatmap.2 (gplots), displaying logCPM of genes related to carbohydrate transport that are differentially expressed in arbuscular mycorrhizal (AM)-colonised root tissues when plant-parasitic nematodes (PPN) are present/absent on the root system. Colour scale reflects logCPM z scores. All data were collected at 5 weeks post inoculation. Treatments are asymbiotic roots, AM-colonised roots, PPN-infected roots, dual PPN + AM fungi colonised roots, and split-root treatments that contained AM fungi in one compartment and PPN in the opposing compartment. $N = 4$, statistical analysis through DESeq2 pairwise comparisons, FDR $p < 0.01$.

infected roots of the same host (Fig. 4A). Despite this, there were no differences in palmitic acid content of extraradical AM fungal hyphae, whether PPN were absent or present within the same or a different root compartment (Fig. 4B).

Roots colonised by AM fungi or infected with PPN contained greater concentrations of glucose and fructose (Figs. 4C, E, S8). Roots colonised by AM fungi accumulated more fructose and glucose than PPN-colonised roots of the same host plant (Fig. 4C, E). In contrast, the concentration of both hexoses in the extraradical AM fungal hyphae was reduced when PPN were present in the root system, regardless of the compartment they were inoculated into (Fig. 4D, F).

### Expression of AM sugar importer genes is independent of the magnitude of C transfer

RNA reads were aligned to a reference AM fungal genome to analyse the AM fungal response to PPN infection of their hosts. Gene expression profiles of AM fungi in these samples clustered towards each treatment (Fig. S9). The expression of AM fungal genes involved in the assimilation of host monosaccharides were stable when PPN were simultaneously infecting the same host, compared to AM fungi in the absence of PPN (Fig. S10). Additionally, the expression of detected AM fungal genes related to the transport of nutrients was also stable when

PPN were infecting the same host, compared to AM fungi-only roots (Fig. S11).

### Conserved shift in host carbon resources transferred to AM fungi

To test the specificity of the hexose-based regulatory mechanism for host carbon outlay to AM fungal symbionts across pests and host plant species, we inoculated aphids (*Myzus persicae*) onto AM-colonised *S. tuberosum* and *Medicago truncatula* hosts. The degree of colonisation of both plant hosts by AM fungi did not differ in the presence of aphids (Fig. S12). However, aphid infestation of above-ground tissues reduced the biomass of both shoots and roots of AM-colonised *S. tuberosum* and *M. truncatula* hosts (Fig. S13).

As per our previous experiments, we assessed the amount of glucose, fructose, and C16:0 palmitic acid within roots and extraradical AM fungal hyphae of co-colonised hosts. As observed with PPN, the palmitic acid content of *S. tuberosum* and *M. truncatula* roots was unaffected by the presence of foliar aphids (Fig. 5A). However, aphid feeding significantly reduced the glucose and fructose content of below-ground tissues (Fig. 5C, E). This pattern was consistent across *S. tuberosum* and *M. truncatula* host plants, even though the amount of each compound detected in the roots of the different species varied.

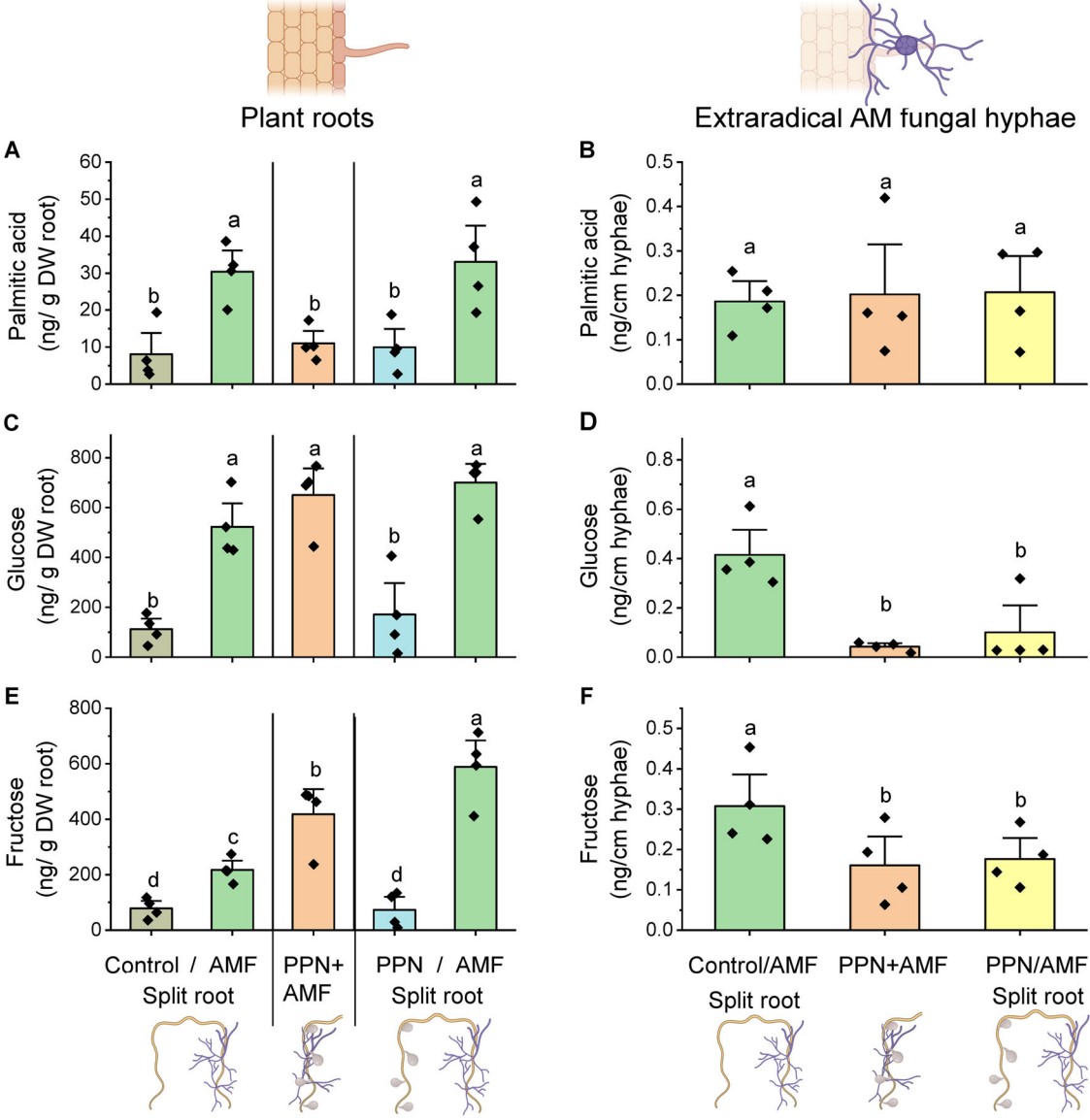

**Fig. 4 | Relative concentrations of C16:0 palmitic acid, fructose and glucose within plant root tissues and extraradical arburscular mycorrhizal fungal (AMF) hyphae in presence and absence of plant-parasitic nematodes (PPN).** Concentrations of (**A**, **B**) palmitic acid, (**C**, **D**) fructose, and (**E**, **F**) glucose in plant roots and extraradical AM fungal hyphae, within each treatment (control = no symbiont, '/' = a single root system split between the two stated treatments,

'+' = both symbionts together on the same roots). All data were collected at 5 weeks post inoculation. Bars represent four biological replicates with the standard error of the mean. Different letters denote significance (**A**, **C** & **E**: linear mixed-effects models were applied to account for the non-independency of split roots belonging to the same plant, Tukey's Honest Significant Difference test, $p < 0.05$. **B**, **D**, **F**: one-way ANOVA, Tukey's Honest Significant Difference test, $p < 0.05$).

The palmitic acid content of extraradical AM fungal hyphae did not vary when the fungus was colonising aphid-free or aphid-infested hosts (Fig. 5B). Furthermore, the concentration of palmitic acid within AM fungal hyphae was consistent across different host plant species whether aphids were present or not. However, the glucose and fructose content of the extraradical AM fungal hyphae was dramatically reduced when plant hosts were subjected to aphid herbivory, compared to aphid-free controls (Fig. 5D, F). Again, this trend was consistent across both *S. tuberosum* and *M. truncatula* hosts.

## Discussion

Our data reveal a mechanism by which plants and AM fungi maintain mutualistic symbioses during concurrent infection of the host plant by parasites. In nature, plants associate with multiple symbionts simultaneously, and these interactions span the entire symbiotic continuum, ranging from mutualistic interactions where both parties

benefit from associating with each other to parasitic interactions where the host incurs a penalty inflicted by the symbiont. Using a split-root experimental design, we compared the amounts and types of carbon compounds transported to AM fungal hyphae by the plant host in the presence and absence of competing symbionts that represent an exogenous sink for plant carbon resources.

In previous experiments[2], we observed a dramatic decrease in plant carbon allocation to AM symbionts in the presence of PPN while AM-mediated transfer of soil nutrients (N and P) was largely maintained. This finding is in line with other observations of carbon-for-nutrient exchange dynamics in plant-AM fungal symbioses in the presence of aphids[19,20]. Given the obligately biotrophic nature of AM fungi, this combination of observations raises the following hypotheses. Firstly, although AM fungi received much less host C when PPN were present, they may have met the minimum amount of plant carbon required to maintain AM fungal mycelia and maintain nutrient

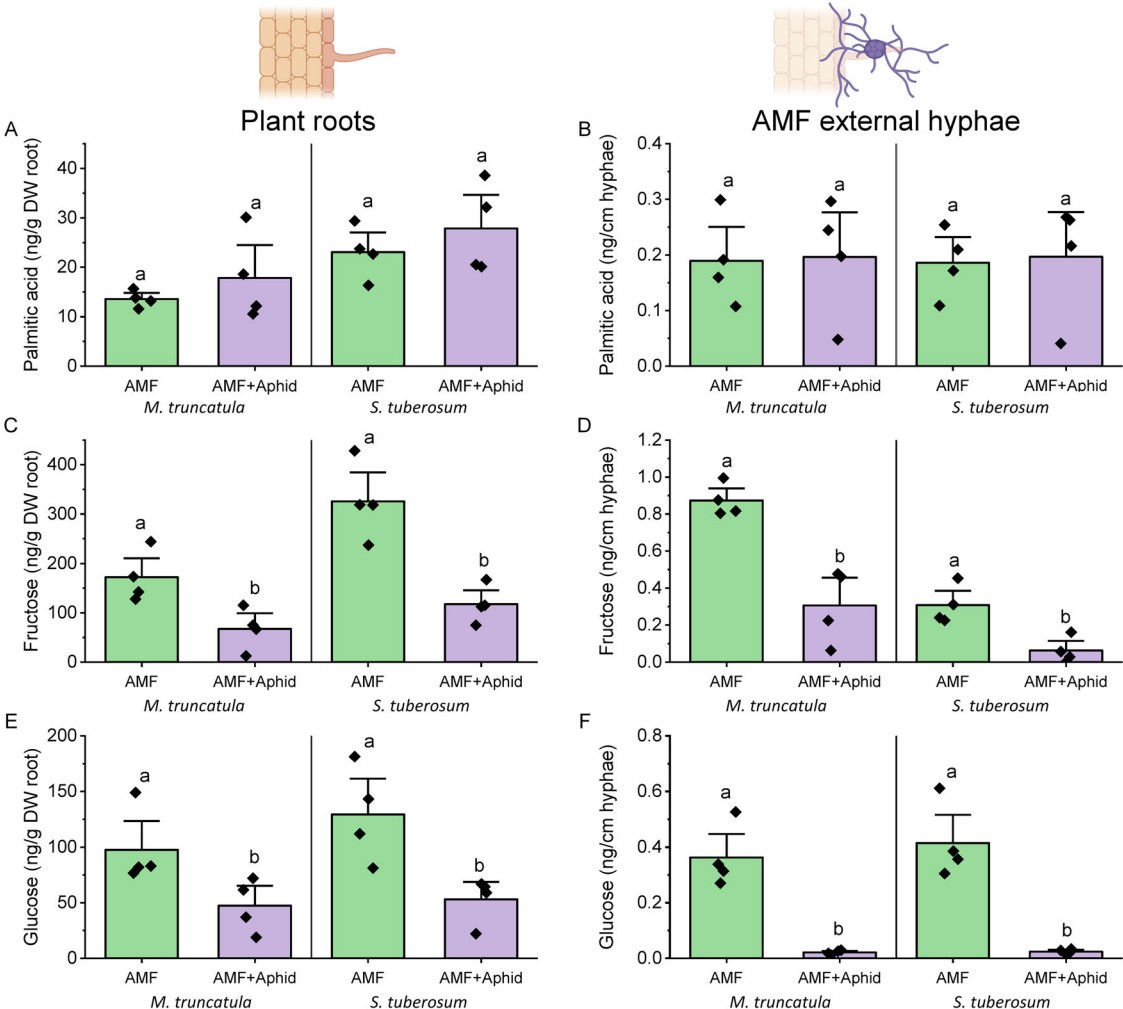

**Fig. 5 | Relative amounts of 16:0 palmitic acid, fructose and glucose within roots of *M. truncatula* and *S. tuberosum* hosts and associated extraradical arbuscular mycorrhizal fungal (AMF) hyphae, in the presence and absence of aphids.** Data show the mean palmitic acid (**A**, **B**), fructose (**C**, **D**) and glucose (**E**, **F**) content of plant roots and extraradical AMF hyphae respectively, within each treatment. All data were collected at 5 weeks post inoculation with AM fungi.

Treatments investigate the impact of foliar-feeding aphids on the concentration of stated compounds within the roots and their associated AMF partners. Bars represent four biological replicates with the standard error of the mean. Different letters denote significance within plot panels, within single plant species (one-way ANOVA, Tukey's Honest Significant Difference test, $p < 0.05$).

flow to the host plant. Secondly, the experimental design may have driven AM fungi into forming, and/or maintaining, partnerships with a sub-optimal, parasitised host with fewer resources available through lack of available host choice. Thirdly, host plants may have withheld photosynthates to all symbionts to reduce the carbon cost of parasitism and better support plant defence responses, with a degree of control over the regulation being dependent on symbiont identity.

In our experiments, roots colonised by either PPN, AM fungi, or both simultaneously, accumulated more plant photosynthates than asymbiotic roots, supporting the idea that sink strength may play a pivotal role in the allocation of host resources in single-symbiont scenarios[28]. Furthermore, transcriptomics analysis indicated that several genes encoding carbohydrate transporters were upregulated in roots colonised by AM fungi, compared to asymbiotic roots from the same host. AM fungi induce changes in host gene expression which increase the synthesis of sucrose in foliar tissues[29], which may then increase the concentration of sucrose breakdown products (glucose and fructose) in colonised root tissue[30]. When both PPN and AM fungi were introduced to plant roots within split-root systems, host carbon was preferentially allocated to AM fungal-colonised roots compared to PPN-infected roots in the same host. Imaging of these roots confirmed

that [14]C-labelled photosynthates were not uniformly distributed across the roots, with regions of the root system accumulating more [14]C than others (Fig. 2B). This may correlate with fungal colonisation 'hotspots', in line with previous observations[31,32]. If the host plant was merely directing carbon resources to avoid PPN-infected roots and thus avoid losses to parasitism, we might have expected asymbiotic roots in the split-root systems to receive most of the plant-fixed carbon resources, however, this was not the case (Fig. 2). Instead, plants appear to actively prioritise resourcing roots that contain beneficial symbionts over parasitic symbionts. Despite this, increased C in AM fungal-colonised roots does not seem to translate to increased C in extraradical AM fungal hyphae. Furthermore, the mechanisms underpinning preferential plant allocation of carbon to sites of beneficial symbiosis and limitation of losses to sites of parasitism are unknown. It is worth noting that PPN still received a substantial amount of carbon-based resources even though the bulk of these compounds were partitioned to mutualist-interacting tissues, potentially indicating that carbon is not limiting for these parasites in this scenario.

AM fungi obtain carbon from their hosts in the form of monosaccharides such as glucose and fructose[33], and fatty acids such as palmitic acid (C16:0[6,7]). We quantified the accumulation of these

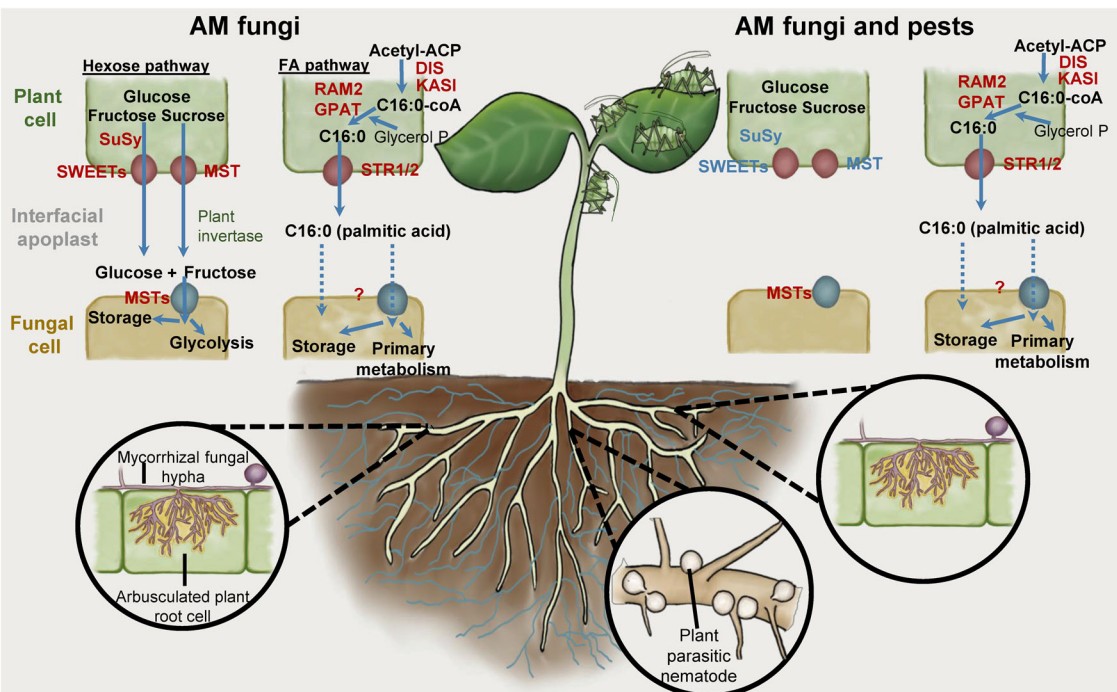

**Fig. 6 | Hypothesised mechanism of carbon transport and allocation from plants to arbuscular mycorrhizal fungi.** Illustrated mechanism underpinning hexose and fatty acid biosynthesis and transport in the absence (left hand side) and presence (right hand side) of pests (aphids or plant-parasitic nematodes) that compete for host carbon resources. Critical gene upregulation upon colonisation by AM fungi indicated using red lettering, and decreased gene expression in AM fungal-colonised tissues in the presence of pests is shown in blue. MST Mono-Saccharide Transporter, SWEETs Sugars Will Eventually be ExporTed, SuSy Sucrose Synthase, DIS/KASI ß-Ketoacyl-Acyl carrier protein Synthase I, RAM2 Glycerol-3-phosphate acyl transferase, STR1/2 Heterodimeric ABCG transporters. "?" represents hypothesised routes of entry for plant-derived C16:0 fatty acids into fungal cells via protein transporters or diffusion[18,63,64].

compounds in host plant roots and their subsequent transfer to the extraradical mycelium of the AM fungus. In our experiments, all three compounds were enriched in AM fungal-colonised roots compared to PPN-infected roots from the same host, again demonstrating enrichment of host C resources in AMF-colonised tissue. Although the total C delivered to AM fungal hyphae from their host decreased when PPN were co-infecting the host, we found that the amount of palmitic acid (C16:0) present within the extraradical fungal mycelium remained consistent across co-symbiont treatments (Fig. 6). The lipid biosynthesis-delivery pathway from host to AM fungi is regulated by the mycorrhizal-induced transcription factor RAM1, which controls downstream enzymes such as FatM, KASI, and GPAT[6,7,34]. Synthesised fatty acids are predicted to be transferred into the peri-arbuscular space by the membrane localised STR1/2 for fungal uptake[24]. We found that whilst genes within the lipid biosynthesis/transport pathway were upregulated in mycorrhizal roots, in agreement with previous studies[6], PPN infection in either root compartment of split-root systems did not impact their expression, corresponding with our finding that palmitic acid (C16:0) delivery to AM fungi was maintained across treatments (Fig. 6). Interestingly, when both symbionts were locally interacting with root tissue (PPN + AMF), we saw a reduction in root palmitic acid, potentially from local priming of defence responses. However, this did not appear to impact the acquisition of this compound by AM fungal partners. Interestingly, *RAM2*, previously described as a mycorrhiza-specific gene[17], was upregulated in PCN-infected roots and potentially relates to the accumulation of palmitic acid in these tissues. This may reflect overlapping mechanisms by which by mutualistic AM partners and parasitic nematodes obtain host resources.

In contrast, the amount of glucose and fructose detected in extraradical AM fungal hyphae was dramatically reduced when PPN were present on the host, even if PPN were restricted to different root tissues than the AM fungi (Fig. 6). *SWEET* genes are important in plant-AM fungi interactions and their upregulation is described as key for the export of carbon to AM fungi[25,26]. Consistent with Manck-Götzenberger & Requena (2016), we found upregulation of *SWEET1b* and *7a* in AM fungi compared to asymbiotic plant roots. Expression of the glucose transporter *SWEET1b* increases upon mycorrhizal colonisation in several host species[26,35–37] with overexpression correlating with enhanced fungal growth indicating its importance in AM fungal nutrition[25]. The loss of AM fungal-regulated *SWEET1b* has been suggested to contribute to the non-mycorrhizal status of some plants including *Arabidopsis thaliana*[9,25]. Interestingly, the upregulation of this *SWEET* gene, along with other mycorrhizal-induced *SWEET*s, was repressed in AM fungal-colonised roots of plants with PPN inoculated into the non-AM root compartment. Roots with reduced expression of these genes exchanged less glucose and fructose with their fungal partners, indicating that mycorrhizal-regulated genes may also respond to biotic stresses, helping to provide greater control over the fate of plant carbohydrates. Expression of *SWEET1b* was also greater in PPN-infected tissues, indicating that as well as being mycorrhizal-induced, parasites may also affect the expression of this gene. It is worth noting that *SWEET* genes most likely serve additional purposes in plants beyond transporting hexoses to AM fungi. They could potentially contribute towards plant processes (e.g. root growth or additional rhizosphere interactions) or for biosynthesis of lipids that may then be utilised by the host or transported to AM partners. Additionally, the movement of hexoses towards AM-colonised roots but apparently not transferred to fungal symbionts indicates distinct transportation pathways/genes for these two processes. There may be local interactions from both PPN and AM fungi impacting *SWEET* gene expression, such as for *SWEET7a,* which was upregulated in both AM fungi-only roots as well as roots colonised concurrently by AM fungi and PPN, but not in the AM-colonised split-root tissues.

In addition to *SWEET*s, we found the induction of a sucrose synthase in mycorrhizal roots that cleaves sucrose into glucose and fructose (Fig. 6), potentially providing monosaccharide molecules for transfer to the fungus. When the orthologue of this gene in *M. truncatula* is knocked down, fungal colonisation is impaired[38] demonstrating the importance of pathways and not individual genes in AM fungal nutrition. The transcriptional profile of these genes is consistent with *SWEET12d* and *12e*, which are Clade 3 genes and predicted to transport sucrose towards colonisation sites in *M. truncatula*[39]. Sucrose transport and cleavage into glucose and fructose increases the monosaccharide concentrations in arbuscocytes, presumably increasing diffusion along gradients into the peri-arbuscular space through hexose transporters or SWEETs. The upregulation of these genes in AM roots is diminished if PPN is present in the root system (Fig. 6), indicating additional genes may regulate AM fungal-colonised roots depending on interacting biotic factors.

Our findings suggest that while PPN infection causes the host plant to redirect carbon resources towards AM fungal-colonised roots, it also restricts the export of sugars while maintaining the transfer of fatty acids. To explore the potential ubiquity of this regulatory mechanism, we inoculated aphids onto AM fungal-colonised host plants (*S. tuberosum* and *M. truncatula*). These experiments elicited a similarly dramatic reduction in hexose supply to AM fungal extraradical hyphae when aphids were present on the host and maintenance of palmitic acid supply (Fig. 6), suggesting this mechanism of resource allocation may be widespread across plant hosts and parasitic interactions. In similar experiments to those conducted here, Charters et al. observed accumulation of total C in roots of aphid-infested wheat (*Triticum aestivum*). We recorded a similar amount of palmitic acid yet a reduced amount of fructose and glucose in the roots of aphid-infested *S. tuberosum* and *M. truncatula* plants, supportive of this being a widespread response. As specific compound abundance was not analysed in Charters et al., it is also possible—even likely—that other carbon-containing molecules are also involved throughout the host plant in response to aphid herbivory, highlighting the need for further investigations to identify which specific compounds are affected.

In our experiments, extraradical AM fungal hyphae contained ~0.2 ng/cm hypha of palmitic acid, compared to 0.4 and 0.3 ng/cm hypha of fructose and glucose, respectively. As such, a reduction in hexose supply from hosts undergoing concurrent parasitism likely has substantial implications on the total amount of plant carbon obtained by AM fungi. The relatively small quantities of carbon that AM fungi receive when a parasitic symbiont is also interacting with the host[19,21] may be almost entirely derived from plant supply of palmitic acid. It is worth noting that in previous studies plants were exposed to carbon label for a relatively short time (e.g., 24 hr)[19,21,40–43]. This may be sufficient for integration of the labelled carbon into carbohydrate compounds, but it may be insufficient for the label to become integrated into the FA synthesis pathway and transferred to AM fungi, potentially due to the amount and localisation of host sugars, impacting their availability as precursor molecules for lipid synthesis[44]. Thus, data from relatively short labelling periods[19,21,40–43] may better represent sugar-based carbon movement rather than the full complement of carbon-based compounds involved in plant-to-AM fungi transfers. An inherent difficulty in interpreting the results of isotope tracing experiments is that the data represent a snap-shot of the metabolic state of the observed tissue. For compounds that incorporate the tracer label, such as fatty acids and sugars, which may undergo metabolic modification, it is difficult to identify which form host-derived compounds take before and after metabolism within the fungus. It is possible that the presence of plant-parasitic nematodes may stimulate the fungus to metabolise sugars at a greater rate to maintain and build wider mycelial networks, allowing fungal hyphae to explore a greater volume of soil for potential new hosts.

Fungal transporters, including MSTs (monosaccharide transporters), facilitate the movement of plant hexoses out of the peri-arbuscular space and into fungal structures[5]. These can transport a wide range of sugar substrates, potentially representing a mechanism by which AM fungi may scavenge a wide array of resources from the host[5,11]. In our study, we found consistent expression of AM fungi monosaccharide transporters, despite the reduction in both the flow of host sugars to the AM fungi and the expression of related host genes. Although reduction of *MST2* expression impairs mycorrhizal formation[5], it does not seem responsive to the amount of resources it receives and is consistently expressed between our treatments, along with other MSTs (Fig. 6). Overall, this suggests a plant-led regulation of its carbohydrate resources with apparently relatively passive AM fungal symbionts that may take as much as their host permits.

The assimilation of varied carbon sources from their hosts is likely to give AM fungi an advantage over competing organisms, facilitating AM fungal survival when certain carbon compounds are restricted by the plant host. In the multi-symbiont scenarios explored here, maintained transfer of FA to AM fungi may ensure fungal survival due to the higher energy content of FAs compared to monosaccharides[44], which the plant may retain in order to promote defence responses or to offset losses to antagonistic symbionts. By limiting sugar outlay, but preserving lipid supply to AM fungal partners, host plants ensure the viability and functionality of mutualistic AM fungal symbionts, thereby maintaining the benefits of AM-mediated soil nutrient assimilation and provision of pest tolerance[21], while simultaneously minimising the carbon costs associated with parasitism. Therefore, it is likely that C-for-P exchange between plant host-AM fungi may be dependent on specific compounds (such as fatty acids) rather than bulk carbon flow. The finely balanced mechanism of control over plant carbon allocation to AM fungi revealed here highlights the strategic adaptability of both plants and AM fungi. This adaptability enables both to thrive in competitive environments and thereby ensure a sustainable and mutually beneficial relationship is maintained even in the face of other biotic pressures vying for plant carbon resources.

## Methods
### Experimental setup and growth of split-root plants
Single chits from *Solanum tuberosum* cv. Désirée were planted in 10 cm diameter pots containing heat-sterilised sand:topsoil (50:50 RHS Silver Sand:Bailey's Norfolk Topsoil, nutritional content provided in Bell et al.[45]). Plants were grown for ~10 days, until there was adequate root growth for transplanting. At this point, the plant was removed from the soil and replanted in a split-root pot. This consisted of 2 × 20 cm pots side by side with a 2 cm × 2 cm section removed from the tops of both. The removed sections were lined up and the two pots were secured together. The potato plants were then placed in this central zone, with half of the roots in each pot. The small size of the central zone minimised the risk of crossover between pots whilst allowing each separate root system to grow into each individual pot. Symbiont counts post-harvest confirmed the efficacy of this setup. These pots were then inoculated with either 30 g of a commercially available inoculum of the AM fungus *Rhizophagus irregularis* (PlantWorks Limited, UK) or 50 cysts of a plant-parasitic nematode (PPN; *Globodera pallida*, population Lindley). Pots without AM fungi (asymbiotic controls and PPN-only) contained blank AM fungal inoculum, comprising sterilised commercial inoculum, to take into account the effects of the carrier agent. The experimental design included pots with (i) no symbionts in either pot, (ii) AM fungi on one side and no symbiont on the other, (iii) PPN on one side and no symbiont on the other, (iv) both PPN and AMF inoculated onto both sides, or (v) PPN inoculated on one side and AM fungi inoculated on the opposite side of the same host plant split-root system (Table S1). Pots were filled with heat-sterilised sand:topsoil (50:50) and randomised for lay-out in a controlled, containment glasshouse (18 °C/16 h day length, watered every other day).

Each AM fungi-containing pot contained 4 PVC cores that were 26 mm diameter and 85 mm length (Barkston Ltd, UK) that were lined with 35 μm pore nylon mesh. The mesh was fine enough to exclude roots but permit access to AM fungal hyphae[46]. The cores were filled with the same substrate as the pots. A single core in each pot was rotated every other day to sever the hyphal connection between core and plant roots. The remaining core in these pots was kept static to preserve the AM fungal hyphal links. After 5 weeks of growth plants were either harvested for biomass and symbiont counts, or $^{14}$C-labelled for resource-tracking as described below.

The photochemical activity of photosystem II characterised by $F_v/F_m$ (Opti-Sciences, OS-30p+ Chlorophyll Fluorometer) was recorded as a functional indicator of plant photosynthetic efficiency and plant stress[47,48], before harvest. Leaves of similar size from the canopy of each plant were shaded for approximately 20 min prior to measuring.

*S. tuberosum* and *Medicago truncatula* wild-type plants were also setup for inoculation with aphids and AM fungi. *S. tuberosum* were grown as above. *M. truncatula* seeds were sown in compost and then potted on after adequate root growth ~1 week post-germination. Both plant species were grown in single 25 cm pots of sterilised sand:topsoil (50:50) with AM fungal inoculum, as described above, with 4 replicates of each plant species. After three weeks of growth in the final pot, 10 apterous *Myzus persicae* aphids (clone O) were transferred from culture *S. tuberosum* plants to experimental plants. After two weeks plants were harvested for biomass and mass spectrometry analysis, below.

### $^{14}$C labelling of host resources
Five weeks after shoot emergence, the above-ground tissues of plants were enclosed within airtight labelling chambers (Polybags Ltd, London, UK). At the beginning of the photoperiod, $^{14}CO_2$ was liberated into the chamber by adding 2 ml 10% lactic acid to 28 μl $^{14}$C-sodium bicarbonate (1 MBq). One ml of labelled chamber headspace gas was sampled immediately using a hypodermic syringe and every 1.5 h to monitor the drawdown of $^{14}CO_2$ by plants. Gas samples were injected into separate gas-evacuated 20 ml scintillation vials containing equal volumes (i.e., 10 ml) of the liquid scintillants Carbo-Sorb and Permafluor scintillation cocktail (PerkinElmer, Beaconsfield, UK) and quantified through liquid scintillation counting (Packard Tri-carbon 3100 TR, Isotech, Chesterfield, UK).

### Harvesting and preparation of material
Plants were left in situ until a maximum of $^{14}$C flux was detected belowground (24 h post-labelling), at which point 4 mL of 2 M KOH was injected into receptacles inside each airtight chamber to capture the remaining $^{14}CO_2$ gas. Soil cores were then removed from pots, and the soil and plant material were separated into shoots, roots, and tubers as well as soil from the pot, rotated, and static cores. All individual components were weighed fresh. Subsamples were stored at −20 °C until they were freeze-dried (CoolSafe 55-4, LaboGene, Allerød, Denmark). Dry weight measurements of each component were taken, before being analysed for total P and $^{14}$C.

### Quantification of fungal colonisation of roots and soils
Fresh root subsamples were cleaned with tap water for quantification of AM root length colonisation. Roots were stained using the "ink and vinegar" staining method[49]. Root samples were first washed in water and then cleared in 10% KOH (w/v) in a 90 °C water bath for 20 min, and AM fungal structures stained with ink and vinegar (5% Pelikan Brilliant Black, 5% acetic acid, 90% dH2O)[49] in a 90 °C water bath for 15 min. Roots were de-stained in 1% acetic acid and mounted on microscope slides using polyvinyl lacto-glycerol (16.6 g polyvinyl alcohol powder, 10 mL glycerol, 100 mL lactic acid, 100 mL dH$_2$O). Assessment of % root length colonisation was made using the magnified intersection methodology (minimum of 150 intersects per pot, ×200 magnification)[50]. AM fungal hyphae were extracted from 4–5 g of

bulk soil in 500 mL H$_2$O, from which 10 ml was filtered through a cellulose nitrate membrane filter (47 mm diameter, 0.45 um pore size; Sartorius, Göttingen, Germany) and stained with ink and vinegar. Filter papers were then cut in half and mounted on microscope slides using polyvinyl lacto-glycerol and oven-dried at 65 °C for an hour. AM hyphal lengths per pot were calculated using the gridline-intersection methodology (50 fields of view per half filter paper, ×200 magnification)[51]. If any control, asymbiotic replicates contained AM colonisation they were removed from the study.

### Quantification of potato cyst nematode infection
Potato cyst nematode cysts were extracted from the roots at the end of the experiment by vigorously shaking the roots/soil and using Fenwick's (1940) method. This data was used alongside root dry weight data to provide cysts root$^{-1}$. These recovered cysts were opened and the number of eggs was counted and expressed as eggs cyst$^{-1}$. Eighty nematode cysts per pot were used as a single replicate for $^{14}$C analysis.

### Quantification of host-fixed C within the system
To quantify $^{14}$C activity in plant, soil and nematodes, respective samples underwent oxidation (Model 307 Packard Sample Oxidiser, Isotech, Chesterfield, UK) followed by liquid scintillation (Packard Tri-Carb 4910TR, PerkinElmer, Beaconsfield, UK). 10–100 mg of freeze-dried sample for plant tissue, soil and nematodes from each pot was weighed in triplicate into Combusto-cones (PerkinElmer, Beaconsfield, UK). $^{14}$C within each material was measured following sample oxidation (Model 307 Packard Sample Oxidiser, Isotech, Chesterfield, UK) whereby released $CO_2$ gets trapped in 10 mL of the liquid scintillant CarboSorb and mixed with 10 mL Permafluor (PerkinElmer). Radioactivity within samples was then quantified through liquid scintillation counting (Packard Tri-Carb 4910TR, PerkinElmer, Beaconsfield, UK). The total C fixed by plants (i.e., $^{12}CO_2$ and $^{14}CO_2$) and transferred to their fungal partner or assimilated by nematodes was calculated by quantifying the total $CO_2$ volume and content mass in the airtight labelling chamber and the proportion of the supplied $^{14}CO_2$ label that was photosynthetically fixed by plants[19,21,23]. To calculate AM fungal C, the rotated core values (i.e., soil and no connected hyphae) were subtracted from the static core (i.e., soil and connected hyphae) to determine the C within the hypha, accommodating for soil C. To determine the proportion of C in the PPN population, data were collected from the 80 sampled cysts and scaled up based on the number of cysts recovered per g of soil, to represent total PPN C.

### Autoradiograph imaging
Prior to freeze-drying and $^{14}$C quantification, the root system of PPN/AMF split-root plant was directly imaged in an electronic autoradiography phosphor imager (Cyclone Plus, PerkinElmer, Beaconsfield, UK) using standard parameters. Only the root system was imaged due to the intense radiolabel presence in foliar tissues, due to foliar fixation of $^{14}$C.

### Determining total P content of shoots
Total P content of plant material was determined using an adapted method[52,53]. Approximately 0.15 mL and 0.2 ml of sulphuric acid digest solutions were added to separate cuvettes with 0.5 ml ammonium molybdate, 0.2 mL of 0.1 M L-ascorbic acid, and 0.2 ml 3.44 M sodium hydroxide. Solutions were made up to 3.8 ml with dH$_2$0. Sample optical density was recorded after 45 mins at 822 nm using a spectrophotometer (Jenway 6300, Staffordshire, UK). A 10 mg/ml standard P solution was used to produce a standard curve against which total sample P mg was calculated.

### Mass spectrometry analysis of host resources
Plants were setup and total roots were freeze-dried as described above. External AM hyphae were extracted from 200 g of total bulk soil.

Specifically, soil was suspended in 1 L H$_2$O and the central vortex of water, containing hyphal strands, was removed via pipetting (~10 ml). This volume was then spun down (3500 rpm 5 mins) to result in the hyphal pellet within a 1.5 ml Eppendorf. These pellets were confirmed to contain AM fungal hyphae via staining with ink-vinegar, as described earlier in this manuscript. Once confirmed, the hyphae were counted and freeze-dried, as described earlier in this manuscript. The hyphal lengths per g of soils provided a normalisation factor for subsequent metabolomics. Twenty mg of freeze-dried plant root material and AM hyphae were subjected to a 30 min ultrasound-assisted extraction, using a mixture of ethyl acetate and methanol (35:65 v/v) at room temperature. The samples were finally centrifuged at 3200 × $g$ for 3 min. These supernatants were freeze-dried and reconstituted in 100 μL 90 AcN/5% water and vortexed for 30 sec, sonicated for 2 min and centrifuged at 9500 × $g$ for 20 min.

**Reversed-phase LC-MS of palmitic acid.** 5 μl of each sample or palmitic acid standards in 90% acetonitrile was injected into a Vanquish LC system (Thermo Scientific, UK) using a flow rate of 0.25 mL min$^{-1}$. The analytical column was Aquity UPLC CSH C18 column (1.7 um particle size, 100 mm × 2.1 mm, Waters, Manchester, UK) held at 50 °C. Starting mobile phase composition was 40% solvent B (0.1% formic acid in 95% acetonitrile/5% IPA) in A (0.1% formic acid and 80% acetonitrile in water) increasing to 54% B over 5 minutes. For column wash and equilibration the %B was increased 70–99%B over 3 minutes, held at 95% B for 2 minutes then returned to 40% B for 5 mins. Column eluant was directed in to an Orbitrap Exploris 240 mass spectrometer (ThermoFisher Scientific, UK) and ionised using electrospray ionisation in negative polarity at 2500 V. Mass measurement used full scan mode with resolution of 120,000, a m/z range of 50–500. The maximum injection time was set automatically by the software. Samples and standards were analysed in triplicate and Tracefinder 5.1 (ThermoFisher Scientific, UK) was used to construct the calibration curve and determine peak areas of palmitic acid signals in the samples. Calibration standard concentrations analysed were 1 ng, 10 ng, 100 ng, 1 μg and 10 μg. Final data were normalised per cm of hyphae per g of starting material.

**HILIC amide LC-MS of glucose and fructose.** 5 μl of each sample or the glucose and fructose standard mixture in 90% acetonitrile was injected into a Vanquish LC system (ThermoFisher Scientific, UK) using a flow rate of 0.25 mL min$^{-1}$. The analytical column was Accucore-150-Amide-HILIC (2.6 μm particle size, 150 mm × 2.1 mm, ThermoFisher Scientific, UK) held at 35 °C. Starting mobile phase composition was 90% solvent B (0.1% ammonium acetate in acetonitrile) in A (0.1% ammonium acetate and water) decreasing to 60% B over 5 minutes and held constant for 4 minutes before being re-equilibrated to 90% B after 2 minutes. Column eluant was eluted into an Orbitrap Exploris 240 mass spectrometer (ThermoFisher Scientific, UK) and ionised using electrospray ionisation in negative polarity at 2500 V. Mass measurement used full scan mode resolution of 120,000, a m/z range of 5–500. The maximum injection time was set automatically by the software. Samples and standards were analysed in triplicate and Tracefinder 5.1 (ThermoFisher Scientific, UK) was used to construct the calibration curve and determine peak areas of hexose signals in the samples. Calibration standard concentrations analysed were 1 ng, 10 ng, 100 ng, 1 μg and 10 μg. Final data were normalised per cm of hyphae per g of starting material.

## Transcriptomics

Split-root treatments were set up as described above with *S. tuberosum*, with 5 replicates of each. Five weeks after planting, roots were harvested, briefly washed to remove soil and then frozen at −80 °C. Roots were then ground in pestle and mortar in liquid nitrogen and total RNA was prepared from nematode samples using an EZNA Plant RNA Kit (Omega Bio-tek) according to the manufacturer's protocol including DNase treatment. RNA quality and quantity were confirmed by Nanodrop spectrophotometer (ThermoFisher Scientific, UK) and Bioanalyser (Agilent Technologies, USA). Samples with an RNA Integrity Number (RIN) greater than 8 were considered sufficient quality for library construction and sequencing. RNA samples were sent for Standard RNA sequencing (Genewiz, UK) for an estimated 30 million reads per sample. Raw reads were quality trimmed with Trimmomatic to remove leading and trailing bases of lower quality. Trimmed reads were mapped to the host plant genome[54] with HISAT2. The unmapped reads were then taken for alignment to both the plant-parasitic nematode (*Globodera pallida*[55]) and AM fungus (*Rhizophagus irregularis*[13]), with alignment statistics provided (Table S2). Reads were then counted with HTSeq and statistically analysed via DESeq2 and EdgeR. Genes with *FDR* < 0.05 and over twofold log$_2$ change were classed as differentially expressed (DEG). These were annotated via BLAST and Interpro searches via OmicsBox Blast2GO[56,57]. Specific SWEET nomenclature[58] and DIS/KASI[6] were annotated as previously described. DEGs annotated as potential resource transporters were then filtered for further analysis.

## qPCR confirmation of AM fungal gene expression

qPCR was performed to confirm the expression profiles of AM fungal genes derived from the above transcriptomics due to the relatively fewer reads returned for AM fungi compared to hosts. First-strand cDNA was synthesised from 500 ng RNA using SuperScript II reverse transcriptase (Invitrogen, UK) and Oligo(dT)17 primer (500 μg/ml) following the manufacturer's protocol. Analysis of gene expression was carried out using quantitative reverse transcription (qRT) PCR with Brilliant III Ultra-Fast SYBR Green Master Mix (Agilent Technologies, CA, USA). Cycle conditions were 95 °C for 30 s and subsequently 40 cycles of 5 s at 95 °C and 10 s at 60 °C. The transcriptome data and matched genome sequence were used to design primers for each AM fungal gene (Table S3 for primer sequences). Each primer pair had an amplification efficiency of 90–105%. The 2$^{(-\Delta\Delta Ct)}$ method was used to calculate relative expression between AM fungi in PPN ± hosts, for three biological replicates each with three technical replicates.

## Statistical analysis

Statistics were carried out using OriginPro[59]. Data were tested for normality using Shapiro−Wilk test. For comparing multiple, independent treatments, data were analysed by one-way ANOVA. Upon *p* < 0.05, post hoc Tukey's Honest Significant Difference tests were run to identify statistical differences between the treatments. Barplot error bars show the standard error of the mean, and the specific statistical comparisons are described in the respective figure legends. "Control" and "PPN + AMF" treatments contain the same inoculum in either half of the split-root system, therefore, root data were averaged between both halves of the treatment to then give a single value that may be compared to other split-root treatments.

For split-root data, a mixed-effects model was constructed in R v4.03 using the lme4 package[60], with "treatment" specified as a fixed effect and "individual host plant" as a random effect to account for the non-independency of split roots belonging to the same plant. Significance was tested using Satterwaite's method with the lmerTest package[61], with further ANOVA and post-hoc analysis through the emmeans package[62] using Tukeys method. Whilst useful for viewing the profiles of a large number of genes, an inherent issue with heatmaps is understanding which data are statistically different from each other, rather than overall trends. Therefore, the Fig. 3 heatmap was generated to combine the expression profile of numerous genes alongside their statistical differences between the different treatments. The colour scale indicates the expression profiles (i.e., higher or lower gene expression for that gene across treatments), and the letters denote the significance of these differences.

**Reporting summary**

Further information on research design is available in the Nature Portfolio Reporting Summary linked to this article.

## Data availability

The data generated during transcriptomic analyses of symbiont-interacting root tissues of *Solanum tuberosum* are available in the SRA database under PRJNA1005093. The datasets generated during other experiments from the current study are available from the corresponding author on reasonable request.

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

## Acknowledgements

We gratefully acknowledge funding from the Leverhulme Trust (RPG-2019-162) to K.J.F. and P.E.U. C.A.B. is supported by a BBSRC Discovery Fellowship (BB/X009823/1) and Michael Beverley Innovation Fellowship. K.J.F. is supported by a BBSRC Translational Fellowship (BB/M026825/1) and E.R.C. CoG MYCOREV (865225). E.M. is supported by a BBSRC White Rose DTP PhD studentship. K.J.F. and E.M. are grateful to the De Laszlo Foundation for Ph.D. student support. The Orbitrap Exploris 240 and Vanguish UPLC was funded by the NIHR (NIHR200633). We thank Dr. Catherine Lilley for her constructive feedback throughout the entirety of this project, Joanna Koch-Paszkowski for technical assistance with autoradiography, and Dr. Chris Bass and Victoria Mallot for the supply of aphid cultures. For the purpose of open access, the author has applied a Creative Commons Attribution (CC BY) licence to any Author Accepted Manuscript version arising from this submission. Barplot pictograms were obtained from BioRender.com 2023.

## Author contributions

C.A.B. and E.M. performed experimental work. J.A. conducted LC-MS experiemnts and analyses. C.A.B. analysed data and wrote the manuscript. P.E.U. and K.J.F. supervised and edited the manuscript. Schematic Fig. 6 was drawn by K.J.F. All authors read and approved the final manuscript.

## Competing interests

The authors declare no competing interests.
