## [Peer Review File · Nature Communications]

Phytophagy impacts the quality and quantity of plant carbon resources acquired by mutualistic arbuscular mycorrhizal fungiREVIEWER COMMENTS

Reviewer #1 (Remarks to the Author):

Summary: A substantial amount of fixed carbon in plants is transferred to both mutualistic and parasitic symbionts, however, how carbon allocation to these symbionts is regulated by the plant is not known. In addition, whether and how plants can prioritise carbon allocation to one interaction partner over the other when both are interacting with a plant at the same time is poorly understood. In this study, the authors investigate how carbon allocation changes in situations when either only one interaction partner or both partners are present in the same or distant root tissues. The authors use both metabolic as well as transcriptional analyses and find that when phytophagous pests (such as nematodes) and mutualistic arbuscular mycorrhizal fungi are present at the same time, the plant preferentially allocates carbon to AMF compared to nematodes. The authors also describe that in these situations, the amount of transferred lipids remains the same, however, the amount of transferred sugars decreases. Understanding the mechanisms of carbon allocation to mutualistic and parasitic microbes is important, particularly considering the potential impact of carbon loss on crop productivity and yield. The finding that sugar transfer to AMF is reduced when nematodes are present is intriguing, and the use of a split-root system to investigate these questions is a good approach that allows investigating many relevant questions. However, some important questions remain unanswered, and there are several inconsistencies in the presented data that are not addressed satisfactorily by the authors. Considering these issues (listed in detail below), I do not think that the conclusions are supported sufficiently by the data, and substantial changes/additional experiments are required to confirm the hypotheses/suggested mechanism underlying the regulation of carbon allocation to mutualistic and parasitic symbionts in this manuscript.

Major points:

1. Missing data to support conclusions about a mechanism for preferential carbon allocation to AMF: While in figure 2, the data for control/PPN is shown, providing evidence for a reduction of carbon allocation to PPN infected roots when AMF are present in a split-root system, none of the subsequent analyses (transcriptional, metabolic) include the control/PPN data. This means that nothing can be concluded on the regulation of carbon allocation to PPN when AMF are present compared to a situation when only PPN are present in the root. This would be important to understand the differences between carbon allocation to AMF and to parasitic symbionts. This is particularly relevant in Figure 4, where palmitic acid, glucose, and fructose contents are shown for all scenarios except for a nematode only system. Without this control, we don't know whether sugars accumulate to a similar amount in roots with PPN as with AMF, and whether this amount is reduced in a PPN/AMF split root system. Transcriptionally, it would have been relevant to test whether sugar transporters are similarly regulated during PPN infection, and whether they are equally down-regulated in a multi-interaction situation as during AMF infection. Based on labelled carbon, we would assume that sugars do accumulate in roots infected by PPN, as the total root carbon content increases (Figure 2). It therefore seems plausible that at least some SWEET transporters are up-regulated in roots to enhance allocation of shoot-fixed carbon to the root. However, without these analyses, it is difficult to confirm the biological relevance of a reduction in sugar allocation to AMF when nematodes are present in a split-root system. A minor comment here: In Figure 3 and 4, PPN are drawn for the AMF only treatment. This is confusing, and also highlights that this data (PPN only) is missing in these figures.
2. Transcriptional data in Figure 3: The gene expression levels are difficult to compare between treatments. It is unclear what the scale shows - the figure legend states that the scale is relative, going from "higher expression" to "lower expression". This implies fold changes. How does this work? Does this mean down- or up-regulation of gene expression, and if so, compared to what? A more conventional way of presenting gene expression would be more appropriate and would allow the reader to interpret the data and judge the quality of the data. A heatmap for overview would be best, and plotting the normalized counts for important genes where differences between treatments are detected is required. This would allow the reader to compare gene expression levels and variability between replicates. The current figure does not provide enough information to draw conclusions on changes in gene expression levels between treatments.
3. Palmitic acid measurements: While amounts of palmitic acid are high in plant roots, the amounts seem to be much lower when measuring fungal hyphae, and standard deviations appear very large. With such small amounts and high variability between the data points, I question

whether the authors are able to detect significant differences in palmitic acid transferred to AM fungi. The authors show that palmitic acid contents in PPN+AMF roots are reduced to control levels, however, in extraradical hyphae the same amounts of palmitic acids are present. This could be another sign for insufficient sensitivity in these measurements. However, one of the main conclusions of this paper is based on these measurements. This is the case for both nematode and aphid treatments. A more appropriate experiment would have been to trace carbon using ^{14}C labelled CO_2 and test how much of the label ends up in fungal hyphae, as either palmitic acid or sugars. The sensitivity would have likely been higher, and differences in the amounts of palmitic acid transferred would have been clearer.

4. In figure 3, SWEET1b is down-regulated in the PPN/AMF system compared to AMF only, and the authors suggest that this is the reason (along with downregulation of other SWEETs) for a reduction in sugar transfer to AMF. However, a reduction in sugar transfer is also apparent in PPN+AMF, based on sugar content in external hyphae, yet SWEET1b expression is not reduced. This inconsistency needs to be addressed.

5. The authors compare carbon allocation to AMF when either nematodes or aphids are present at the same time, and conclude from the data that a reduction in sugar allocation to AMF is a conserved mechanism. However, these are two very different pests, with nematodes infecting the roots, and aphids feeding on the shoots. This will have a major impact on fixed carbon and its allocation from shoots to roots. In the case of aphids, shoot carbon will be significantly reduced, meaning that less carbon is available to be allocated to AM fungi overall. In the case of nematodes, carbon allocation from shoots to roots could differ between AMF and nematode compartments. Unfortunately, no $^{14}\text{CO}_2$ experiments are shown for the aphid experiments, and so nothing can be concluded on the total amount of carbon that is allocated to roots with and without aphids.

5. Fungal colonization levels: Although authors quantify total colonization and state that colonization levels are the same across all treatments (Figure S2), it is unclear whether the infection structures, particularly arbuscules, remain the same and functional when parasites are present. Gene expression analysis showed that StPT4 expression is reduced in PPN+AMF and PPN/AMF treatments - this indicates that the symbiosis might not be as functional as when only AMF are present, and that arbuscules might senesce prematurely (Figure S5). WGA staining and microscopy images of arbuscules would help in determining whether the symbiosis is functional in all treatments. Furthermore, In Figure S3, no data is provided on plant shoot total P in an AMF only infected plant, making it difficult to conclude whether as much phosphate is provided by AMF with additional parasites present.

Minor points:

1. It would be helpful to describe in the results section when (how many weeks after inoculation) colonization, metabolites, gene expression was measured.

4. In the discussion, SWEET transporters are all described as transferring sugars to AMF. However, different SWEETs might have different functions. For example, some SWEETs might be involved in transferring sugars to arbuscule containing cells for lipid biosynthesis. It needs to be made clear that not all SWEETs will be directly involved in sugar transfer to fungal hyphae at the peri-arbuscular membrane, and that sugar allocation to roots might serve to provide substrates for lipid biosynthesis, not just for sugar provision to AM fungi.

Reviewer #2 (Remarks to the Author):

This manuscript explores whether plant antagonists (plant pathogenic nematodes and aphids) influence the allocation of different forms of carbon to an AM fungal beneficial partner (*Rhizophagus irregularis*). The study used a large number of measurements (isotopes, metabolomics, transcriptomics, and qPCR) to track the carbon molecules being exchanged between plants and the different organisms. Together these measurements create a clear picture of the exchange occurring between organisms. I would have liked to know the proportion of carbon forms in the root that were located within the AM fungal hyphae, but I know of no methods for measuring this variable. The manuscript was well-written and my comments are directed at improving the writing clarity or providing missing information in the methods and results.

1) It was interesting that there was no difference among the treatments in AM fungal colonization.

- However, I was concerned as to why AM fungal colonization of the control was not included. Sure this is necessary to include to demonstrate that the treatments worked?
- 2) I found the frequent use of the word symbiont confusing, primarily as it was not always clear to which symbiont the text was referring. Especially line 23 on page 3 as here nematodes and aphids are referred to as "competing symbionts" for the first time. Perhaps give symbionts adjectives (e.g., beneficial or antagonistic), and consider defining the word symbiont as not all readers recognize that symbionts can be something other than mutualistic.
 - 3) Line 10 on page 4: is "phytophagous" not typically used to describe aboveground plant antagonists? If so, is it appropriate to include nematodes within this group?
 - 4) Line 16 on page 4: I was intrigued by the positive influence of AM fungi on potato biomass, as this is in contrast to multiple papers by Karley and Bennett where multiple potato species showed no response to AM fungi.
 - 5) After reading the methods I was under the impression that there was no treatment combining AM fungi and nematodes in the same pot. In particular, the wording of treatment iv in lines 18-19 on page 25 did not clearly state that both organisms were in both halves of the pot. Also, based on Table S1 there are five treatments (including the control). One way to clarify this would be to include Table S1 in the main document or to refine the text to more clearly describe the treatments.
 - 6) Line 5 on page 6: Change to read "More host-fixed carbon was..."
 - 7) Line 2 on page 8: Change to read "...distinct gene expression profile..."
 - 8) Lines 11-17 on page 8: These two sentences read as if they are discussing the same thing (carbohydrate transport) but coming to different conclusions (reduced versus up-regulated).
 - 9) Lines 7-8 on page 12: This is the first mention of unsuccessful transcriptomic analyses—something I would have expected to see in either the previous section or in the methods.
 - 10) Lines 18-20 on page 14: Here the authors argue that AM fungi would have formed partnerships with sub-optimal hosts already infected with an antagonist. However, most AM fungi form partnerships with plants prior to the plant encountering any kind of antagonist. Therefore, instead of forming a relationship these fungi are being forced to maintain a relationship they might not otherwise. While the distinction seems small, the implications are not as the fungi will have already invested in infrastructure in a given host.
 - 11) Lines 5-6 on page 15: I did not buy the argument that competition between nematodes and AM fungi for carbon was an example of 'reciprocal rewards'. The plant knows the difference between AM fungi and nematodes and their purpose in the roots, and therefore carbon moving to nematodes is being "stolen" whereas carbon moving towards AM fungi is plant allocated. This is like comparing apples and oranges, and the comparison is not valid.
 - 12) Line 16 on page 16: Change to read "...root tissues than the AM fungi (Fig 6)."
 - 13) Line 17 on page 17: Change to read "...may regulate the AM fungal..."
 - 14) Line 20 on page 17: Change to read "... towards AM fungal-colonised roots..."
 - 15) Line 24 on page 17: Change to read "...when aphids were present..."
 - 16) Line 4 on page 25: Please include details on how the sand:soil mix was sterilized as well as source and nutrient profile of the topsoil.
 - 17) Lines 10-11 on page 26: *Myzus persicae* were cultured on *Solanum tuberosum* and then transferred to *Solanum tuberosum* and *Medicago trunculata* plants. Given that *M. persicae* tend to adapt to the host they feed on and transfer between hosts creates a bottleneck that significantly reduces survival, would this not influence aphid performance on *M. trunculata* versus *S. tuberosum*?
 - 18) Lines 2-3 on page 28: Was the entire plant (aboveground and belowground) imaged? The image in Figure 2 suggests that just the belowground was imaged. Could that be clarified here? Also, was only one plant imaged? If so, why only one?
 - 19) Lines 5-12 on page 28: This section is missing the following citation: Irving, G.C.J. and M.J. McLaughlin, A rapid and simple field test for phosphorus in Olsen and Bray No 1 extracts of soil. *Communications in Soil Science and Plant Analysis*, 1990. 21(19-20): p. 2245-2255.
 - 20) Lines 21 on page 28: Add a full stop (.) after "metabolomics" and before "Twenty".
 - 21) Line 21 on page 30: Why was OriginPro used instead of R here?
 - 22) Line 28 on page 30: What is meant by "single, single-pot value"?
 - 23) Line 30 on page 30: Include the version of R used and the package(s) used and their citations.
 - 24) Line 1 on page 31: Change to read: "...rather than overall trends."

Reviewer #3 (Remarks to the Author):

Bell et al. explored the concomitant interactions which plants have with their (typically) beneficial fungal symbionts, AM fungi, along with plant parasitic nematodes or piercing/sucking insect herbivore.

With the fanfare and interest that research into carbon allocation from host plant to fungi receives, I am always disappointed we still have such little understanding of how invertebrate herbivores/parasites alter the AM fungal-host plant transfer of resources, and here we have a significant paper that really does advance our understanding of these interactions. I wish I had conducted this work myself.

It is particularly striking that the authors were able to find the same overall trend when comparing the sugar and lipid to the AM fungi when hosts are simultaneously challenged by the PPN and the insect herbivore! These are two very different modes of plant-feeding antagonists. The work has significant implications for our understanding of these complex interactions. Does that have meaning for AM-induced plant defences? While the reduction in carbon is almost totally a reduction in fats, what might this mean for communities of AM fungi that inhabit roots? Do we think there would be differences among AM fungal taxa in terms of their capacity to maintain symbiosis when sugar supply is so limited as a result of herbivore/parasite attack? I'm not expecting the authors to address these, but just want to highlight the important research questions that will build from this work.

I have some specific points below:

P3, L8-10: I assume this is not to be consistent across AM fungal taxa? We know that AM fungi differ in their growth habits in terms of the amount of hyphal biomass they produce and where and how that is allocated between soil and within the root. So presumably the minimum would potentially be taxon-group specific.

P3, L18-20: I'm not sure I follow the second part of this statement so easily. What do you mean about the 'otherwise cannot be regulated'? Maybe I'm just being dense, but what is this referring to?

P3, L22-24: I read this section first and couldn't follow. It then became clear in the next paragraph that it was because of your reference to all parties here (AM fungi, aphid, PPN) as symbionts. While of course, you are totally correct! It is not common to refer to aphids (and maybe less so PPN) as symbionts. Thus, perhaps here, and throughout, is there a more intuitive way to refer to these groups? Again, I know you are technically correct, but to maximise simplicity and understandability. Have a think about this terminology application.

P4, L18: Interesting that AM fungi did not enhance photosynthesis? Was this expected?

P4, L24: Yes, this seems to be quite consistent outcome for PPN-AMF interactions.

P4, L31-33: I think this is really interesting. Considering your findings then, is the whole paradigm of C-P exchange potentially not so much based on the sugars, but rather the fats? Consider this transfer seems to be quite stable.

Figure 2: Could the figure say something else on the y-axis? At first, I thought, wow! What are these roots made of if not carbon before I realised it was the fixed carbon. Maybe this can be clearer on the figure 2A itself?

P8, L16-25: This is an impactful finding in my opinion. The consistency of the lipid transfer, yet the carbohydrates are more subject to oscillation depending on whether the plant has aphid/PPN or not.

Figure 3: I do wonder if there is a way the authors could make it more obvious the differences? In my own annotations, I found myself connecting the two 'circles' to highlight differences. This would get messy indeed if it were done for all. And I'm not sure I have a better suggestion than what you

have already, but maybe give it some thought.

P10, L11: Does $N = 4$ have sufficient power in these contexts?

P14, L2-4: I think this first sentence could be stronger. From a fungal perspective, they are looking to maintain their carbon gain, and seem to be able to do so via lipid transfer.

P18, L20: But presumably the total carbon to the AM fungi is still much less, right? So can we say there is much 'growth'? Probably more simply metabolic maintenance? I'm not sure, but I don't think the fungi would be as growing or 'productive' with only the FA's available to them. Presumably, they would like some of the carbohydrates as well.

P25, L15: Would the sterilization of the inoculum have altered the nutrient profile at all? If by heat then this might 'release' some nutrients which would not have otherwise been available to the plant.

P27, L10-11: I may have missed it, but do we then know the total hyphal biomass per pot? Or totally hyphal length per pot? This would be quite informative.

Reviewer #4 (Remarks to the Author):

Dear Editor

We would like to thank both yourself and the reviewers for the positive and supportive comments towards our manuscript and the suggestions for improvement. Here we provide a revised version of the manuscript and below, describe the revisions made in response to the reviewers' comments (*red*). A second manuscript document has the changes highlighted in red, as requested.

Reviewer #1

Understanding the mechanisms of carbon allocation to mutualistic and parasitic microbes is important, particularly considering the potential impact of carbon loss on crop productivity and yield. The finding that sugar transfer to AMF is reduced when nematodes are present is intriguing, and the use of a split-root system to investigate these questions is a good approach that allows investigating many relevant questions. However, some important questions remain unanswered, and there are several inconsistencies in the presented data that are not addressed satisfactorily by the authors. Considering these issues (listed in detail below), I do not think that the conclusions are supported sufficiently by the data, and substantial changes/additional experiments are required to confirm the hypotheses/suggested mechanism underlying the regulation of carbon allocation to mutualistic and parasitic symbionts in this manuscript.

We thank the Reviewer for appreciating the importance of this study. We would like to respectfully note that where the Reviewer notes an “inconsistency” or “unsupported hypotheses” that these are additional hypotheses raised by the Reviewer that this study did not set out to test. As such, these hypotheses were unsupported in the original submitted version of the manuscript, however, we acknowledge their value and have addressed them here or in the manuscript where appropriate (see detailed comments below).

1. Missing data to support conclusions about a mechanism for preferential carbon allocation to PPN: While in figure 2, the data for control/PPN is shown, providing evidence for a reduction of carbon allocation to PPN infected roots when AMF are present in a split-root system, none of the subsequent analyses (transcriptional, metabolic) include the control/PPN data. This means that nothing can be concluded on the regulation of carbon allocation to PPN when AMF are present compared to a situation when only PPN are present in the root

Our manuscript does not aim or attempt to conclude on plant regulation of carbon allocation to PPN. This is an element introduced here by the reviewer and was not an aim of our study; neither of the other two reviewers understood our manuscript to investigate carbon flow to PPN. Despite this, we have added these additional datasets to the manuscript (Fig 3, Fig S5, S6, S8), as requested by the reviewer (these data were collected alongside those presented within the original submission but omitted for conciseness). We display gene expression for PPN only root tissue in the main text as per the reviewer's suggestion, and use this as the rationale (P9L19) for presenting treatments only related to AM fungi in Figure 4. The PPN-related LC-MS data is presented in Figure S8 for the reviewer. We agree with the reviewer that these data are interesting from a plant-PPN perspective, however our manuscript focuses on understanding carbon flow to AM fungi and we prefer to maintain this focus in the main manuscript.

This would be important to understand the differences between carbon allocation to AMF and to parasitic symbionts. This is particularly relevant in Figure 4, where palmitic acid, glucose, and fructose contents are shown for all scenarios except for a nematode only system. Without this control, we don't know whether sugars accumulate to a similar amount in roots with PPN as with AMF, and whether this amount is reduced in a PPN/AMF split root system.

Our manuscript does not make conclusions regarding sugar accumulation in PPN-infected roots but focuses on the regulation and flow of resources towards AM fungi. However, we have included these data in Fig S8 to satisfy the reviewer's request. Here, the data indicate that hexoses and palmitic acid compounds accumulate in nematode-infected roots, but less so if AM fungi are colonising the split-root, as shown in Fig 2A. The data suggest that reduction of these compounds in PPN-infected tissues is not detrimental or limiting to their performance (as described in the text, P15L10).

Transcriptionally, it would have been relevant to test whether sugar transporters are similarly regulated during PPN infection, and whether they are equally down-regulated in a multi-interaction situation as during AMF infection. Based on labelled carbon, we would assume that sugars do accumulate in roots infected by PPN, as the total root carbon content increases (Figure 2). It therefore seems plausible that at least some SWEET transporters are up-regulated in roots to enhance allocation of shoot-fixed carbon to the root. However, without these analyses, it is difficult to confirm the biological relevance of a reduction in sugar allocation to AMF when nematodes are present in a split-root system.

We have included hexose-related transcriptomic and metabolomic data in PPN only conditions. These data indicate that certain SWEET genes are upregulated in PPN-infected roots. We present the expression of AM fungi-induced sugar-related genes in PPN infected tissue in Fig 3. The biological relevance of sugar flow to AMF when PPN are present is discussed on P19-L1-16.

A minor comment here: In Figure 3 and 4, PPN are drawn for the AMF only treatment. This is confusing, and also highlights that this data (PPN only) is missing in these figures.

We thank the Reviewer for this comment and have corrected the pictograms in Fig 4, whilst Fig 3 has been revised in line with other comments.

2. Transcriptional data in Figure 3: ...A more conventional way of presenting gene expression would be more appropriate and would allow the reader to interpret the data and judge the quality of the data. A heatmap for overview would be best, and plotting the normalized counts for important genes where differences between treatments are detected is required.

We have combined comments from Reviewer's 1 and 3 to change Fig 3 to aid clarity. We now present expression profiles for genes that are related to sugar transport that are differentially expressed in AMF-only roots compared to in AM-colonised roots with distal PPN infection. This indicates the distal impact of PPN on AM fungal-root interactions. We thank both reviewers for their constructive feedback. The main text figure is the recommended heatmap whilst the normalised counts for both sugar-related and fatty acid biosynthesis/transport-related genes (which are not significantly different between these two treatments) are provided in Fig S5, S6.

3. Palmitic acid measurements: While amounts of palmitic acid are high in plant roots, the amounts seem to be much lower when measuring fungal hyphae, and standard deviations appear very large.

The units used to express palmitic acid abundance in roots vs. hyphae are necessarily very different; per g root whilst for AM fungi it is expressed as per cm hyphae. We do not expect the amounts of any compound to be equally, or more, abundant in AM hyphae compared to root tissues given the direction of flow from plant to fungi. Contrary to the Reviewers comment, if the two values were indeed similar then we would expect an error in the experimental methods.

With such small amounts and high variability between the data points, I question whether the authors are able to detect significant differences in palmitic acid transferred to AM fungi. The authors show that palmitic acid contents in PPN+AMF roots are reduced to control levels, however, in extraradical hyphae the same amounts of palmitic acids are present. This could be another sign for insufficient sensitivity in these measurements. However, one of the main conclusions of this paper is based on these measurements. This is the case for both nematode and aphid treatments. A more appropriate experiment would have been to trace carbon using ¹⁴C labelled CO₂ and test how much of the label ends up in fungal hyphae, as either palmitic acid or sugars. The sensitivity would have likely been higher, and differences in the amounts of palmitic acid transferred would have been clearer.

LC-MS can detect very low quantities of palmitic acid, as shown by the standard curve provided in the figure below. The data here have been log transformed to demonstrate the ability of the method to detect, in this instance, 1ng of palmitic acid. In our experimental runs we extracted hyphae from approximately 200 cm of hyphae, detected the total compound abundance and then divided this by the hyphal lengths to generate ng/cm hyphae displayed in Figure 4 in our manuscript. Based on the data from Fig 4 in the manuscript, a minimum of approximately 15 ng of palmitic acid was detected in fungal hyphae, which is well over the detection limits of ~1ng suggested by the standard curve.

Standard curve of 1ng, 10ng, 100ng, 1000ng of palmitic acid detected via LC-MS.

The reviewer's suggestion to trace a $^{14}\text{CO}_2$ label into fungal hyphae and use this to identify compound identity would certainly be a novel approach, however it is technically intractable as the ^{14}C (half-life >5,000 years) would contaminate the instrument and be prohibitively expensive due to the replacement of instrument components after each run. Combined with the transcriptomic and isotope tracing data, the LC-MS data presented in our manuscript provides compelling evidence that there is a reduction of hexose sugar yet maintenance of palmitic acid flow to extraradical AM fungal hyphae when PPN co-colonise root systems.

4. In figure 3, SWEET1b is down-regulated in the PPN/AMF system compared to AMF only, and the authors suggest that this is the reason (along with downregulation of other SWEETs) for a reduction in sugar transfer to AMF. However, a reduction in sugar transfer is also apparent in PPN+AMF, based on sugar content in external hyphae, yet SWEET1b expression is not reduced. This inconsistency needs to be addressed.

We thank the Reviewer for asking for clarification on this. In joint PPN+AMF tissues we observed changes in gene expression derived from both parasitic and mutualistic interactions. Therefore, rather than an inconsistency, this indicates that as well as being potentially mycorrhizal-regulated, SWEET1b may additionally be regulated by the presence of parasitic nematodes. The extra data provided in Fig S5 suggests that this is true. We have added this interesting point to P17 L7.

5. The authors compare carbon allocation to AMF when either nematodes or aphids are present at the same time, and conclude from the data that a reduction in sugar allocation to AMF is a conserved mechanism. However, these are two very different pests, with nematodes infecting the roots, and aphids feeding on the shoots...

We agree with the reviewer. The stark contrasts between PPN and aphids was the rationale for choosing these pests to investigate conservation of the reduced sugar mechanism.

...This will have a major impact on fixed carbon and its allocation from shoots to roots. In the case of aphids, shoot carbon will be significantly reduced, meaning that less carbon is available to be allocated to AM fungi overall....

We thank the Reviewer for their comment, however this is not true. Shoot carbon is not significantly reduced during aphid feeding at ambient CO_2 levels (Charters *et al.*, 2020, *Current Biology*, 30: 1801-1808).

...In the case of nematodes, carbon allocation from shoots to roots could differ between AMF and nematode compartments. Unfortunately, no $^{14}\text{CO}_2$ experiments are shown for the aphid experiments, and so nothing can be concluded on the total amount of carbon that is allocated to roots with and without aphids.

Here, we show that carbon delivery to AMF and PPN compartments does differ (Figure 2a). However, regarding the reviewer's comment that "nothing can be concluded on the total amount of carbon allocated to roots with and without aphids"; again, this was not something we hypothesised or specifically tested in our experiments with aphids as previous studies have already shown total root carbon is greater in aphid infested plants than in non-infested plants (Charters *et al.*, 2020). In this manuscript we investigated hexose and palmitic acid content of roots and AM external hyphae from +/- aphid treatments. Together, these results are very interesting considering we found similar amounts of palmitic acid yet reduced hexose quantities in roots from aphid infested plants compared to AM only plants and this

may explain the observations regarding AM-carbon in Charters et al (2020). We have added this interesting discussion point to P18L2.

5. Fungal colonization levels: Although authors quantify total colonization and state that colonization levels are the same across all treatments (Figure S2), it is unclear whether the infection structures, particularly arbuscules, remain the same and functional when parasites are present.

As part of the AM fungal colonisation assessment, we noted abundance of each AM structure and have amended Fig S2 to include this additional information. The functionality of this symbiosis is inferred by the response to the comment below, that there are comparable levels of plant shoot P between treatments including AM fungi, and our prior work (cited in the manuscript) has comprehensively quantified the transfer of labelled AM-acquired P into host plants (Bell *et al.*, 2022, *New Phytologist* 234: 269-279). This is a far more accurate assessment of mycorrhizal function than observation and reporting of presence and abundance of fungal structures alone.

Furthermore, In Figure S3, no data is provided on plant shoot total P in an AMF only infected plant, making it difficult to conclude whether as much phosphate is provided by AMF with additional parasites present.

We apologise for the oversight and have included these data for total shoot P in AM only plants in Fig S3 as requested. To conclude, we see no reductions in total shoot P when PPN are present.

1. It would be helpful to describe in the results section when (how many weeks after inoculation) colonization, metabolites, gene expression was measured.

We have copied this information from the Methods section to where relevant within the figure legends.

4. In the discussion, SWEET transporters are all described as transferring sugars to AMF. However, different SWEETs might have different functions. For example, some SWEETs might be involved in transferring sugars to arbuscule containing cells for lipid biosynthesis. It needs to be made clear that not all SWEETs will be directly involved in sugar transfer to fungal hyphae at the peri-arbuscular membrane, and that sugar allocation to roots might serve to provide substrates for lipid biosynthesis, not just for sugar provision to AM fungi.

We have added to P17L9 to include this interesting point.

Reviewer #2 (Remarks to the Author):

We thank the Reviewer for their supportive and positive comments regarding our manuscript.

1) It was interesting that there was no difference among the treatments in AM fungal colonization. However, I was concerned as to why AM fungal colonization of the control was not included. Sure this is necessary to include to demonstrate that the treatments worked?

We routinely measure the colonisation of control roots to check that the sterilised soil and blank inoculum does not yield AM colonisation. If any roots are colonised by AM fungi then we discard these plants from the experiment, confirming our “control” plants were not colonised. We apologise for the oversight in not including this information in our original manuscript and have now added this to the Methods P28L12.

2) I found the frequent use of the word symbiont confusing, primarily as it was not always clear to which symbiont the text was referring. Especially line 23 on page 3 as here nematodes and aphids are referred to as “competing symbionts” for the first time. Perhaps give symbionts adjectives (e.g., beneficial or antagonistic), and consider defining the word symbiont as not all readers recognize that symbionts can be something other than mutualistic.

We agree and have amended this section as well as including a line to P3L16 to clarify that antagonistic organisms are also symbionts.

3) Line 10 on page 4: is “phytophagous” not typically used to describe aboveground plant antagonists? If so, is it appropriate to include nematodes within this group?

Plant-parasitic nematodes are indeed phytophagous due to their ability to feed from plant tissues, and are commonly referred to as phytophagous (e.g. Espada et al., 2015, *Molecular Plant Pathology* 17(2): 286-295). We prefer to retain this terminology but are happy to change it at the discretion of the editor.

4) Line 16 on page 4: I was intrigued by the positive influence of AM fungi on potato biomass, as this is in contrast to multiple papers by Karley and Bennett where multiple potato species showed no response to AM fungi.

This result appears to be variable between publications, however we consistently see a positive impact of AM fungi on potato biomass, in line with other publications (Yang et al., 2020. *J. Plant Nutrition and Soil Science* 183(2): 20-232)

5) After reading the methods I was under the impression that there was no treatment combining AM fungi and nematodes in the same pot. In particular, the wording of treatment iv in lines 18-19 on page 25 did not clearly state that both organisms were in both halves of the pot. Also, based on Table S1 there are five treatments (including the control). One way to clarify this would be to include Table S1 in the main document or to refine the text to more clearly describe the treatments.

We thank the reviewer for this point. We have amended P26L18 to clarify the treatments, and align them with Table S1.

6) Line 5 on page 6: Change to read “More host-fixed carbon was...”

Done.

7) Line 2 on page 8: Change to read "...distinct gene expression profile..."

Done.

8) Lines 11-17 on page 8: These two sentences read as if they are discussing the same thing (carbohydrate transport) but coming to different conclusions (reduced versus up-regulated).

We have amended P8 L13-14 to clarify our findings to the reader.

9) Lines 7-8 on page 12: This is the first mention of unsuccessful transcriptomic analyses—something I would have expected to see in either the previous section or in the methods.

We checked expression profiles of AM gene expression by qPCR to confirm RNAseq data, rather than due to unsuccessful RNAseq. We have added to P31L14 to clarify this.

10) Lines 18-20 on page 14: Here the authors argue that AM fungi would have formed partnerships with sub-optimal hosts already infected with an antagonist. However, most AM fungi form partnerships with plants prior to the plant encountering any kind of antagonist. Therefore, instead of forming a relationship these fungi are being forced to maintain a relationship they might not otherwise. While the distinction seems small, the implications are not as the fungi will have already invested in infrastructure in a given host.

We thank the reviewer for this valuable addition and have included it in P14L18. It is true that AM fungi may colonise hosts before as well as after an antagonist.

11) Lines 5-6 on page 15: I did not buy the argument that competition between nematodes and AM fungi for carbon was an example of 'reciprocal rewards'. The plant knows the difference between AM fungi and nematodes and their purpose in the roots, and therefore carbon moving to nematodes is being "stolen" whereas carbon moving towards AM fungi is plant allocated. This is like comparing apples and oranges, and the comparison is not valid. Not fully sure what the point is here. Can try and change though

We have amended P15 L5-8 to remove reference to the reciprocal rewards theory.

12) Line 16 on page 16: Change to read "...root tissues than the AM fungi (Fig 6)."

Done.

13) Line 17 on page 17: Change to read "...may regulate the AM fungal-..."

Done.

14) Line 20 on page 17: Change to read "... towards AM fungal-colonised roots..."

Done.

15) Line 24 on page 17: Change to read "...when aphids were present..."

Done.

16) Line 4 on page 25: Please include details on how the sand:soil mix was sterilized as well as source and nutrient profile of the topsoil.

We have added information regarding the source and sterilisation procedure for soil mixes. We have referenced Bell et al., 2023 (Bell, C.A., et al. 2023. Sequence of introduction determines the success of contrasting root symbionts and their host.

Applied Soil Ecology. 104733) in this line, which provides the nutrient profile of this soil mixture, which we have used in recently published research.

17) Lines 10-11 on page 26: *Myzus persicae* were cultured on *Solanum tuberosum* and then transferred to *Solanum tuberosum* and *Medicago trunculata* plants. Given that *M. persicae* tend to adapt to the host they feed on and transfer between hosts creates a bottleneck that significantly reduces survival, would this not influence aphid performance on *M. trunculata* versus *S. tuberosum*?

This is a very good point; this species of aphid tends to adapt to the specific host that from which they are feeding, and performs better on that host over time. However, in this study we did not specifically compare data across plant species, rather we compared between AM/non-AM treatments within each individual plant species. As such, aphid performance does not affect our comparisons. We have added to P12 L27 to clarify this in text.

18) Lines 2-3 on page 28: *Was the entire plant (aboveground and belowground) imaged? The image in Figure 2 suggests that just the belowground was imaged. Could that be clarified here? Also, was only one plant imaged? If so, why only one?*

Only below-ground materials were imaged due to foliar fixation of the radiolabelled carbon resulting in intense radiation of above-ground tissues that saturated images when scanned. We have added text to P28L4-7 to clarify this. Multiple plants were imaged, however this representative plant (Fig 2B) was selected to best illustrate the quantitative data collected from a number of plants tested (Fig 2A, C, D) and presented within the same figure. The representative image in Fig 2B also helps to contextualise and illustrate the discussion point on P14L36 that fungal colonisation sites may correlate with carbon “hotspots” (Lerat *et al.* 2003).

19) Lines 5-12 on page 28: *This section is missing the following citation: Irving, G.C.J. and M.J. McLaughlin, A rapid and simple field test for phosphorus in Olsen and Bray No 1 extracts of soil. Communications in Soil Science and Plant Analysis, 1990. 21(19-20): p. 2245-2255.*

We have added this reference to the section.

20) Lines 21 on page 28: *Add a full stop after “metabolomics” and before “Twenty”.*

Done.

21) Line 21 on page 30: *Why was OriginPro used instead of R here? Why not?*

The authors prefer the interface and functionality of OriginPro, therefore use this as default for statistics and plotting. The outputs are not affected by the choice of statistical software package used.

22) Line 28 on page 30: *What is meant by “single, single-pot value”?*

We have amended P30L33 to clarify that in treatments that had the same inoculum on each side of the split roots (i.e. Control and PPN+AMF) we averaged data from both sides to give a single value for that replicate.

23) Line 30 on page 30: *Include the version of R used and the package(s) used and their citations.*

We have added to the packages cited with a citation for “lme4” P30L34.

Reviewer #3 (Remarks to the Author):

We thank Reviewer 3 for their strong, positive comments regarding the significant impact our manuscript will have on advancing understanding of AM-host interactions. We agree that this paper raises important questions for future studies and will shape our knowledge of AM-induced plant defences, how host resource transfer may influence the success of varying taxa, and what the long-term implications may mean for AM communities (from both academic and applied perspectives).

P3, L8-10: I assume this is not to be consistent across AM fungal taxa? We know that AM fungi differ in their growth habits in terms of the amount of hyphal biomass they produce and where and how that is allocated between soil and within the root. So presumably the minimum would potentially be taxon-group specific.

We thank the reviewer and have added to P3L9 to emphasise that there may be AM fungal species effects not tested for here.

P3, L18-20: I'm not sure I follow the second part of this statement so easily. What do you mean about the 'otherwise cannot be regulated'? Maybe I'm just being dense, but what is this referring to?

We thank the Reviewer for this comment and have amended P3L20 to aid clarity.

P3, L22-24: I read this section first and couldn't follow. It then became clear in the next paragraph that it was because of your reference to all parties here (AM fungi, aphid, PPN) as symbionts. While of course, you are totally correct! It is not common to refer to aphids (and maybe less so PPN) as symbionts. Thus, perhaps here, and throughout, is there a more intuitive way to refer to these groups? Again, I know you are technically correct, but to maximise simplicity and understandability. Have a think about this terminology application.

We have amended the specific wording in this paragraph to indicate when we are referring to AM fungi or other competing symbionts. We have further changed subsequent use of this term as per Reviewer 2's advice to label "antagonistic symbionts".

P4, L18: Interesting that AM fungi did not enhance photosynthesis? Was this expected?

We have found AM fungal colonisation doesn't impact FvFm of potato hosts in several independent experiments. This appears to vary between host species and is likely dependent on the host identity or the growth conditions the plants are exposed to.

P4, L24: Yes, this seems to be quite consistent outcome for PPN-AMF interactions.

We agree, it is unfortunate that AM fungi seem to drive increased PPN populations rather than offering "bio-control" properties that have been previously reported.

P4, L31-33: I think this is really interesting. Considering your findings then, is the whole paradigm of C-P exchange potentially not so much based on the sugars, but rather the fats? Consider this transfer seems to be quite stable.

We hypothesise that the paradigm referred to is much more complex than was initially proposed. In our manuscript, we propose that the fatty acids may be provided to AM fungi on a stable, long-term basis compared to the dynamic fluctuations of sugar flows that may vary with biotic perturbations. This hypothesis is

complementary with the reviewer's hypothesis regarding the potentially larger importance of fatty acids and we have included this very interesting point on P19L10-12.

Figure 2: Could the figure say something else on the y-axis? At first, I thought, wow! What are these roots made of if not carbon before I realised it was the fixed carbon. Maybe this can be clearer on the figure 2A itself?

We have changed the y axis accordingly to clarify that we present "recently fixed root carbon". We apologise for this confusion.

P8, L16-25: This is an impactful finding in my opinion. The consistency of the lipid transfer, yet the carbohydrates are more subject to oscillation depending on whether the plant has aphid/PPN or not.

We thank the reviewer for their support and fully agree, we think this is a very exciting finding!

Figure 3: I do wonder if there is a way the authors could make it more obvious the differences? In my own annotations, I found myself connecting the two 'circles' to highlight differences. This would get messy indeed if it were done for all. And I'm not sure I have a better suggestion than what you have already, but maybe give it some thought.

We thank the Reviewer for this comment. In light of this and Reviewer 1's comment, we have altered Fig 3 show only the sugar-transport related genes that are differentially expressed in AMF-colonised roots depending on the co-presence of distal PPN. This allows us to produce a figure that more clearly presents the genes of interest for this manuscript. The fatty acid related genes are presented via a plot of normalised gene counts and given in Fig S6, as requested by Reviewer 1.

P10, L11: Does $N = 4$ have sufficient power in these contexts?

Determining the statistical power of RNAseq experiments is difficult due to the varying number of reads returned per replicate and the varying number of differentially expressed genes identified that each have their own, potentially independent, variance. Not knowing these numbers prior to the experiment prevents robust estimations of power. As per Schurch *et al.* 2016 we used *DESeq2* (as well as checks with *edgeR*) to identify differentially expressed genes with a fold-change >2. This captures the majority of true differential expression (>85% true positive rate) and increasing replicate number only has an effect on detection rate of genes with smaller fold changes (i.e. >0.5), of which did not meet our selected cut-off. (Schurch, N.J., *et al* 2016. How many biological replicates are needed in an RNA-seq experiment and which differential expression tool should you use? *RNA* doi.org10.1261/rna.053959.115)

For additional quality checks we observed that our replicates clustered particularly well within their treatments (Fig S4), which offers assurance of limited variation, and the AM fungal differentially expressed genes were confirmed via qPCR. Finally, following guidance from Schurch *et al.* 2016, RNASeq experiments published recently in *Nature Communications* used fewer replicates than this study with robust results (Siddique *et al*, 2022) which gives us strong confidence that we have sufficient power in our reporting of transcriptomics data to draw the inferences and conclusions drawn in our manuscript.

P14, L2-4: I think this first sentence could be stronger. From a fungal perspective, they are looking to maintain their carbon gain, and seem to be able to do so via lipid transfer.

We have amended this opening sentence (P14L2) to include the fungal perspective.

P18, L20: But presumably the total carbon to the AM fungi is still much less, right? So can we say there is much 'growth'? Probably more simply metabolic maintenance? I'm not sure, but I don't think the fungi would be as growing or 'productive' with only the FA's available to them. Presumably, they would like some of the carbohydrates as well.

We have amended P19L2 to acknowledge this good point that, at the moment in time it was measured, the fungus may not have been growing as readily, and maintenance of fatty acid transfer may ensure fungal survival until monosaccharides are transferred again.

P25, L15: Would the sterilization of the inoculum have altered the nutrient profile at all? If by heat then this might 'release' some nutrients which would not have otherwise been available to the plant.

The blank inoculum was the same substrate/carrier agent, but without the AM fungal addition. This was utilised to account for any nutrients that the carrier agent may possess. The inoculum was not sterilised as such, rather it did not contain the fungal addition.

P27, L10-11: I may have missed it, but do we then know the total hyphal biomass per pot? Or totally hyphal length per pot? This would be quite informative.

In Fig S2 we present hyphal lengths per g dry soil, which did not vary between treatments. Our pots each contained approximately 3kg of soil, and if hyphal length data is extrapolated to the total pot size we still saw no differences between treatments which is interesting; we have referred to this observation in the results (P4 L31-33).

Reviewer #1 (Remarks to the Author):

Dear Editor

We would like to thank both yourself and the reviewers for the positive and supportive comments towards our manuscript and the suggestions for improvement. Here we provide a revised version of the manuscript and below, describe the revisions made in response to the reviewers' comments (*red*). A second manuscript document has the changes highlighted in red, as requested.

Reviewer #1

Understanding the mechanisms of carbon allocation to mutualistic and parasitic microbes is important, particularly considering the potential impact of carbon loss on crop productivity and yield. The finding that sugar transfer to AMF is reduced when nematodes are present is intriguing, and the use of a split-root system to investigate these questions is a good approach that allows investigating many relevant questions. However, some important questions remain unanswered, and there are several inconsistencies in the presented data that are not addressed satisfactorily by the authors. Considering these issues (listed in detail below), I do not think that the conclusions are supported sufficiently by the data, and substantial changes/additional experiments are required to confirm the hypotheses/suggested mechanism underlying the regulation of carbon allocation to mutualistic and parasitic symbionts in this manuscript.

We thank the Reviewer for appreciating the importance of this study. We would like to respectfully note that where the Reviewer notes an “inconsistency” or “unsupported hypotheses” that these are additional hypotheses raised by the Reviewer that this study did not set out to test. As such, these hypotheses were unsupported in the original submitted version of the manuscript, however, we acknowledge their value and have addressed them here or in the manuscript where appropriate (see detailed comments below).

1. Missing data to support conclusions about a mechanism for preferential carbon allocation to AMF: While in figure 2, the data for control/PPN is shown, providing evidence for a reduction of carbon allocation to PPN infected roots when AMF are present in a split-root system, none of the subsequent analyses (transcriptional, metabolic) include the control/PPN data. This means that nothing can be concluded on the regulation of carbon allocation to PPN when AMF are present compared to a situation when only PPN are present in the root

Our manuscript does not aim or attempt to conclude on plant regulation of carbon allocation to PPN. This is an element introduced here by the reviewer and was not an aim of our study; neither of the other two reviewers understood our manuscript to investigate carbon flow to PPN. Despite this, we have added these additional datasets to the manuscript (Fig 3, Fig S5, S6, S8), as requested by the reviewer (these data were collected alongside those presented within the original submission but omitted for conciseness). We display gene expression for PPN only root tissue in the main text as per the reviewer's suggestion, and use this as the rationale (P9L19) for presenting treatments only related to AM fungi in Figure 4. The PPN-related LC-MS data is presented in Figure S8 for the reviewer. We agree with the reviewer that these data are interesting from a plant-PPN perspective, however our manuscript focuses on understanding carbon flow to AM fungi and we prefer to maintain this focus in the main manuscript.

This would be important to understand the differences between carbon allocation to AMF and to parasitic symbionts. This is particularly relevant in Figure 4, where palmitic acid, glucose, and fructose contents are shown for all scenarios except for a nematode only system. Without this control, we don't know whether sugars accumulate to a similar amount in roots with PPN as with AMF, and whether this amount is reduced in a PPN/AMF split root system.

Our manuscript does not make conclusions regarding sugar accumulation in PPN-infected roots but focuses on the regulation and flow of resources towards AM fungi. However, we have included these data in Fig S8 to satisfy the reviewer's request. Here, the data indicate that hexoses and palmitic acid compounds accumulate in nematode-infected roots, but less so if AM fungi are colonising the split-root, as shown in Fig 2A. The data suggest that reduction of these compounds in PPN-infected tissues is not detrimental or limiting to their performance (as described in the text, P15L10).

The addition of the PPN only data allows for a more comprehensive, complete dataset which means all scenarios can be considered. While you have now included these data as requested, there is only minimal mention and discussion of the data in the text, and so it is now up to the reader to interpret what may be happening in the PPN only condition. I understand that you have done this to maintain the focus of the paper on carbon flow to AM fungi, not PPN. If this is the case, then the sections in the text where you refer to "preferential" or "selective" carbon allocation to AM fungi need to be changed because these phrases suggest that you have investigated and discussed both scenarios at a molecular (i.e. transcriptional and metabolic) level. Phrases like this include:

- Page 1, line 19 – 21
- Page 3, line 97-100
- Page 4, line 105-106
- Page 19, line 471-473

Please rephrase these to emphasise that you were interested in carbon allocation to AM fungi only and not the system as a whole.

If you decide to include the additional data, I would suggest that you incorporate Fig S8 into Fig 4 as it is basically a duplicate of the results with the addition of the PPN only sample.

Transcriptionally, it would have been relevant to test whether sugar transporters are similarly regulated during PPN infection, and whether they are equally down-regulated in a multi-interaction situation as during AMF infection. Based on labelled carbon, we would assume that sugars do accumulate in roots infected by PPN, as the total root carbon content increases (Figure 2). It therefore seems plausible that at least some SWEET transporters are up-regulated in roots to enhance allocation of shoot-fixed carbon to the root. However, without these analyses, it is difficult to confirm the biological relevance of a reduction in sugar allocation to AMF when nematodes are present in a split-root system.

We have included hexose-related transcriptomic and metabolomic data in PPN only conditions. These data indicate that certain SWEET genes are upregulated in PPN-infected roots. We present the expression of AM fungi-induced sugar-related genes

in PPN infected tissue in Fig 3. The biological relevance of sugar flow to AMF when PPN are present is discussed on P19-L1-16.

The addition of the PPN only RNA-seq data completes the dataset, makes the AMF results more interesting and substantiates some of the conclusions you have drawn for carbon allocation to AMF.

Your explanation of the biological relevance of sugar flow to AMF is sound but please provide a reference for P19, lines 469 – 471.

A minor comment here: In Figure 3 and 4, PPN are drawn for the AMF only treatment. This is confusing, and also highlights that this data (PPN only) is missing in these figures.

We thank the Reviewer for this comment and have corrected the pictograms in Fig 4, whilst Fig 3 has been revised in line with other comments.

Addressed.

2. Transcriptional data in Figure 3: ...A more conventional way of presenting gene expression would be more appropriate and would allow the reader to interpret the data and judge the quality of the data. A heatmap for overview would be best, and plotting the normalized counts for important genes where differences between treatments are detected is required.

We have combined comments from Reviewer's 1 and 3 to change Fig 3 to aid clarity. We now present expression profiles for genes that are related to sugar transport that are differentially expressed in AMF-only roots compared to in AM-colonised roots with distal PPN infection. This indicates the distal impact of PPN on AM fungal-root interactions. We thank both reviewers for their constructive feedback. The main text figure is the recommended heatmap whilst the normalised counts for both sugar-related and fatty acid biosynthesis/transport-related genes (which are not significantly different between these two treatments) are provided in Fig S5, S6.

Adding the heatmap has improved the presentation of the data, as it is easier to interpret and compare genes between treatments. If you are basing your palmitic acid, glucose and fructose quantification on the fact that hexose-related but not fatty acid-related genes were differentially expressed, I would suggest to include all of these genes in one heatmap to see the whole picture. Please add the genes from Fig S6 (fatty acid-related genes) into the Fig. 3 heatmap.

A few other concerns/requests regarding the transcriptome data:

- Please add the PPN only samples to the Fig. S4 PCA plot for consistency with the rest of the RNA-seq data if you choose to include the data in Figure 3.*
- Please also convert Fig. S7 into a heatmap for consistency with the other data you have presented in the main figures.*
- RAM2, a mycorrhizal specific gene, is going up significantly in all of the treatments compared with the control which is interesting and consistent with the accumulation of palmitic acid in PPN only infected roots. Can you comment on why this may be? Has this been observed previously for PPN*

infected roots? If presenting this data, it is worth mentioning and discussing this result in the manuscript.

3. Palmitic acid measurements: While amounts of palmitic acid are high in plant roots, the amounts seem to be much lower when measuring fungal hyphae, and standard deviations appear very large.

The units used to express palmitic acid abundance in roots vs. hyphae are necessarily very different; per g root whilst for AM fungi it is expressed as per cm hyphae. We do not expect the amounts of any compound to be equally, or more, abundant in AM hyphae compared to root tissues given the direction of flow from plant to fungi. Contrary to the Reviewers comment, if the two values were indeed similar then we would expect an error in the experimental methods.

Apologies if my comment was not clear. I of course did not mean to suggest that you should find more lipids in fungal hyphae than in roots – I simply tried to emphasize that you are looking at very small amounts of palmitic acid and appear to have a very large variability between the data points.

With such small amounts and high variability between the data points, I question whether the authors are able to detect significant differences in palmitic acid transferred to AM fungi. The authors show that palmitic acid contents in PPN+AMF roots are reduced to control levels, however, in extraradical hyphae the same amounts of palmitic acids are present. This could be another sign for insufficient sensitivity in these measurements. However, one of the main conclusions of this paper is based on these measurements. This is the case for both nematode and aphid treatments. A more appropriate experiment would have been to trace carbon using ¹⁴C labelled CO₂ and test how much of the label ends up in fungal hyphae, as either palmitic acid or sugars. The sensitivity would have likely been higher, and differences in the amounts of palmitic acid transferred would have been clearer.

LC-MS can detect very low quantities of palmitic acid, as shown by the standard curve provided in the figure below. The data here have been log transformed to demonstrate the ability of the method to detect, in this instance, 1 ng of palmitic acid. In our experimental runs we extracted hyphae from approximately 200 cm of hyphae, detected the total compound abundance and then divided this by the hyphal lengths to generate ng/cm hyphae displayed in Figure 4 in our manuscript. Based on the data from Fig 4 in the manuscript, a minimum of approximately 15 ng of palmitic acid was detected in fungal hyphae, which is well over the detection limits of ~1 ng suggested by the standard curve.

Standard curve of 1ng, 10ng, 100ng, 1000ng of palmitic acid detected via LC-MS.

The reviewer's suggestion to trace a ¹⁴CO₂ label into fungal hyphae and use this to identify compound identity would certainly be a novel approach, however it is technically intractable as the ¹⁴C (half-life >5,000 years) would contaminate the instrument and be prohibitively expensive due to the replacement of instrument components after each run. Combined with the transcriptomic and isotope tracing data, the LC-MS data presented in our manuscript provides compelling evidence that there is a reduction of hexose sugar yet maintenance of palmitic acid flow to extraradical AM fungal hyphae when PPN co-colonise root systems.

Thank you for providing data on the detection limits of palmitic acid using LC-MS. I do not question that you are able to detect the low amounts of palmitic acids present in fungal hyphae. However, I still have concerns about the large degree of variance you see between replicates. There is a possibility that this variance is why no differences in palmitic acid are detected between the different conditions tested. One option to improve this might be to include more replicates for these measurements.

More generally however, measuring absolute amounts of metabolites only provides a snap shot of the metabolic state of a tissue. A possible interpretation based on the data you provide is that lipids accumulate in plant roots under different conditions, but that these lipids are not actually exported to AM fungi (this is under the assumption that you are not confidently detecting differences in palmitic acid levels between different treatments). The same is true for sugars. Based on the provided data (i.e. lower amounts of sugars in fungal hyphae), it is not possible to distinguish for example between a case where less sugars are exported to AM fungi, as opposed to a situation where the fungus might not break down the lipids it receives from plants into sugars because it uses lipids as longer-term carbon storage and for survival, particularly if overall much less carbon is being transferred to AM fungi, as your carbon tracing data (Figure 2) suggests. It is also possible that you observe lower amounts of sugars in fungal hyphae when PPN are present because the fungus metabolises these sugars more quickly. Overall, you are not directly

measuring flux/transport of metabolites, simply the amount present in different tissues.

The field has struggled to determine the relative amounts of carbon delivered to AM fungi in the form of lipids versus sugars for years. This is due to the technical difficulties associated with carbon labelling. One possible way around this would be to use carbon isotope tracing, as I suggested above. I understand that you do not want to use LC-MS to measure ¹⁴C labelled metabolites due to the long half-life of ¹⁴C. An alternative would be to use ¹³C. Importantly, one would have to do flux measurements, rather than just measuring absolute amounts of metabolites at one time point, to be able to measure export of metabolites from roots to AM fungi.

I would not want to suggest that you include these additional experiments for this publication, as these experiments are time consuming and difficult to do. However, if the current data is the only data you can provide, I strongly suggest that you include a detailed discussion on possible alternative interpretations of the metabolic data. In my opinion, you cannot conclusively show export of certain metabolites from plant roots to AM fungi based on the data you provide. This needs to be reflected in a more careful discussion of the findings.

4. In figure 3, SWEET1b is down-regulated in the PPN/AMF system compared to AMF only, and the authors suggest that this is the reason (along with downregulation of other SWEETs) for a reduction in sugar transfer to AMF. However, a reduction in sugar transfer is also apparent in PPN+AMF, based on sugar content in external hyphae, yet SWEET1b expression is not reduced. This inconsistency needs to be addressed.

We thank the Reviewer for asking for clarification on this. In joint PPN+AMF tissues we observed changes in gene expression derived from both parasitic and mutualistic interactions. Therefore, rather than an inconsistency, this indicates that as well as being potentially mycorrhizal-regulated, SWEET1b may additionally be regulated by the presence of parasitic nematodes. The extra data provided in Fig S5 suggests that this is true. We have added this interesting point to P17 L7.

Addressed.

5. The authors compare carbon allocation to AMF when either nematodes or aphids are present at the same time, and conclude from the data that a reduction in sugar allocation to AMF is a conserved mechanism. However, these are two very different pests, with nematodes infecting the roots, and aphids feeding on the shoots...

We agree with the reviewer. The stark contrasts between PPN and aphids was the rationale for choosing these pests to investigate conservation of the reduced sugar mechanism.

...This will have a major impact on fixed carbon and its allocation from shoots to roots. In the case of aphids, shoot carbon will be significantly reduced, meaning that less carbon is available to be allocated to AM fungi overall....

We thank the Reviewer for their comment, however this is not true. Shoot carbon is not significantly reduced during aphid feeding at ambient CO₂ levels (Charters *et al.*, 2020, *Current Biology*, 30: 1801-1808).

*Thank you for pointing out this reference. As far as I understand, Charters et al. 2020 investigated the effect of carbon allocation to AM fungi during aphid feeding in wheat. Is there evidence that shoot to root carbon allocation dynamics in your tested systems (*M. truncatula* and *S. tuberosum*) with aphids are the same as in wheat? Based on your changes on Page 18 lines 425-434, there do seem to be differences between species – am I interpreting this incorrectly?*

...In the case of nematodes, carbon allocation from shoots to roots could differ between AMF and nematode compartments. Unfortunately, no ¹⁴CO₂ experiments are shown for the aphid experiments, and so nothing can be concluded on the total amount of carbon that is allocated to roots with and without aphids.

Here, we show that carbon delivery to AMF and PPN compartments does differ (Figure 2a). However, regarding the reviewer's comment that "nothing can be concluded on the total amount of carbon allocated to roots with and without aphids"; again, this was not something we hypothesised or specifically tested in our experiments with aphids as previous studies have already shown total root carbon is greater in aphid infested plants than in non-infested plants (Charters *et al.*, 2020). In this manuscript we investigated hexose and palmitic acid content of roots and AM external hyphae from +/- aphid treatments. Together, these results are very interesting considering we found similar amounts of palmitic acid yet reduced hexose quantities in roots from aphid infested plants compared to AM only plants and this may explain the observations regarding AM-carbon in Charters *et al* (2020). We have added this interesting discussion point to P18L2.

I agree that this is a very interesting observation, but would like to reiterate here that the authors need to be careful about conclusions that are drawn from the data provided regarding export of sugars and lipids to AM fungi, as discussed above. If the authors did not set out to test the hypothesis that less carbon is allocated to roots with and without aphids, this has to be made clear in the text (as suggested for comment 1).

5. Fungal colonization levels: Although authors quantify total colonization and state that colonization levels are the same across all treatments (Figure S2), it is unclear whether the infection structures, particularly arbuscules, remain the same and functional when parasites are present.

As part of the AM fungal colonisation assessment, we noted abundance of each AM structure and have amended Fig S2 to include this additional information. The functionality of this symbiosis is inferred by the response to the comment below, that there are comparable levels of plant shoot P between treatments including AM fungi, and our prior work (cited in the manuscript) has comprehensively quantified the transfer of labelled AM-acquired P into host plants (Bell *et al.*, 2022, *New Phytologist* 234: 269-279). This is a far more accurate assessment of mycorrhizal function than observation and reporting of presence and abundance of fungal structures alone.

Addressed, now that the missing data for shoot total P in an AMF only infected plant has been provided.

Furthermore, In Figure S3, no data is provided on plant shoot total P in an AMF only infected plant, making it difficult to conclude whether as much phosphate is provided by AMF with additional parasites present.

We apologise for the oversight and have included these data for total shoot P in AM only plants in Fig S3 as requested. To conclude, we see no reductions in total shoot P when PPN are present.

Addressed.

1. It would be helpful to describe in the results section when (how many weeks after inoculation) colonization, metabolites, gene expression was measured.

We have copied this information from the Methods section to where relevant within the figure legends.

Addressed.

4. In the discussion, SWEET transporters are all described as transferring sugars to AMF. However, different SWEETs might have different functions. For example, some SWEETs might be involved in transferring sugars to arbuscule containing cells for lipid biosynthesis. It needs to be made clear that not all SWEETs will be directly involved in sugar transfer to fungal hyphae at the peri-arbuscular membrane, and that sugar allocation to roots might serve to provide substrates for lipid biosynthesis, not just for sugar provision to AM fungi.

We have added to P17L9 to include this interesting point.

Addressed.

Reviewer #3 (Remarks to the Author):

The authors have addressed the concerns raised during the previous review with commendable thoroughness and clarity. I appreciate that they navigated through the extensive feedback, particularly addressing Reviewer 1's points, which, while intriguing, were tangential to the scope of this study.

I have read through the response to the reviewers and read the revised manuscript. The research presented is impactful and a significant contribution to our understanding of the AM symbiosis and resource exchange dynamics between the fungus and host plant when said host is simultaneously dealing with plant-feeding antagonists. As the authors highlight, this is a set of interactions ubiquitous in nature.

The revised manuscript now stands as a robust piece of scholarship.

Reviewer #4 (Remarks to the Author):

Dear Editor

We thank you for accepting to publish our manuscript. As requested, we have responded to reviewer 1's latest comments (blue) in yellow highlighted text below, and by making the requested changes within our manuscript file.

Reviewer #1

Understanding the mechanisms of carbon allocation to mutualistic and parasitic microbes is important, particularly considering the potential impact of carbon loss on crop productivity and yield. The finding that sugar transfer to AMF is reduced when nematodes are present is intriguing, and the use of a split-root system to investigate these questions is a good approach that allows investigating many relevant questions. However, some important questions remain unanswered, and there are several inconsistencies in the presented data that are not addressed satisfactorily by the authors. Considering these issues (listed in detail below), I do not think that the conclusions are supported sufficiently by the data, and substantial changes/additional experiments are required to confirm the hypotheses/suggested mechanism underlying the regulation of carbon allocation to mutualistic and parasitic symbionts in this manuscript.

We thank the Reviewer for appreciating the importance of this study. We would like to respectfully note that where the Reviewer notes an "inconsistency" or "unsupported hypotheses" that these are additional hypotheses raised by the Reviewer that this study did not set out to test. As such, these hypotheses were unsupported in the original submitted version of the manuscript, however, we acknowledge their value and have addressed them here or in the manuscript where appropriate (see detailed comments below).

1. Missing data to support conclusions about a mechanism for preferential carbon allocation to AMF: While in figure 2, the data for control/PPN is shown, providing evidence for a reduction of carbon allocation to PPN infected roots when AMF are present in a split-root system, none of the subsequent analyses (transcriptional, metabolic) include the control/PPN data. This means that nothing can be concluded on the regulation of carbon allocation to PPN when AMF are present compared to a situation when only PPN are present in the root

Our manuscript does not aim or attempt to conclude on plant regulation of carbon allocation to PPN. This is an element introduced here by the reviewer and was not an aim of our study; neither of the other two reviewers understood our manuscript to investigate carbon flow to PPN. Despite this, we have added these additional datasets to the manuscript (Fig 3, Fig S5, S6, S8), as requested by the reviewer (these data were collected alongside those presented within the original submission but omitted for conciseness). We display gene expression for PPN only root tissue in the main text as per the reviewer's suggestion, and use this as the rationale (P9L19) for presenting treatments only related to AM fungi in Figure 4. The PPN-related LC-MS data is presented in Figure S8 for the reviewer. We agree with the reviewer that these data are interesting from a plant-PPN perspective, however our manuscript focuses on understanding carbon flow to AM fungi and we prefer to maintain this focus in the main manuscript.

This would be important to understand the differences between carbon allocation to AMF and to parasitic symbionts. This is particularly relevant in Figure 4, where palmitic acid, glucose, and fructose contents are shown for all scenarios except for a nematode only system. Without this control, we don't know whether sugars accumulate to a similar amount in roots with PPN as with AMF, and whether this amount is reduced in a PPN/AMF split root system.

Our manuscript does not make conclusions regarding sugar accumulation in PPN-infected roots but focuses on the regulation and flow of resources towards AM fungi. However, we have included these data in Fig S8 to satisfy the reviewer's request. Here, the data indicate that hexoses and palmitic acid compounds accumulate in nematode-infected roots, but less so if AM fungi are colonising the split-root, as shown in Fig 2A. The data suggest that reduction of these compounds in PPN-infected tissues is not detrimental or limiting to their performance (as described in the text, P15L10).

The addition of the PPN only data allows for a more comprehensive, complete dataset which means all scenarios can be considered. While you have now included these data as requested, there is only minimal mention and discussion of the data in the text, and so it is now up to the reader to interpret what may be happening in the PPN only condition. I understand that you have done this to maintain the focus of the paper on carbon flow to AM fungi, not PPN. If this is the case, then the sections in the text where you refer to "preferential" or "selective" carbon allocation to AM fungi need to be changed because these phrases suggest that you have investigated and discussed both scenarios at a molecular (i.e. transcriptional and metabolic) level.

Phrases like this include:

- Page 1, line 19 – 21 **DONE in Editor corrected abstract**
- Page 3, line 97-100 **DONE L92**
- Page 4, line 105-106 **DONE L108**
- Page 19, line 471-473 **DONE L469**

Please rephrase these to emphasise that you were interested in carbon allocation to AM fungi only and not the system as a whole.

If you decide to include the additional data, I would suggest that you incorporate Fig S8 into Fig 4 as it is basically a duplicate of the results with the addition of the PPN only sample.

We have kept Fig S8 rather than combining to save space and keep the main narrative clear.

Transcriptionally, it would have been relevant to test whether sugar transporters are similarly regulated during PPN infection, and whether they are equally down-regulated in a multi-interaction situation as during AMF infection. Based on labelled carbon, we would assume that sugars do accumulate in roots infected by PPN, as the total root carbon content increases (Figure 2). It therefore seems plausible that at least some SWEET transporters are up-regulated in roots to enhance allocation of shoot-fixed carbon to the root. However, without these analyses, it is difficult to confirm the biological relevance of a reduction in sugar allocation to AMF when nematodes are present in a split-root system.

We have included hexose-related transcriptomic and metabolomic data in PPN only conditions. These data indicate that certain SWEET genes are upregulated in PPN-infected roots. We present the expression of AM fungi-induced sugar-related genes

in PPN infected tissue in Fig 3. The biological relevance of sugar flow to AMF when PPN are present is discussed on P19-L1-16.

The addition of the PPN only RNA-seq data completes the dataset, makes the AMF results more interesting and substantiates some of the conclusions you have drawn for carbon allocation to AMF.

Your explanation of the biological relevance of sugar flow to AMF is sound but please provide a reference for P19, lines 469 - 471.

We have amended this line to indicate that it is our hypothesis rather than a referenced statement.

A minor comment here: In Figure 3 and 4, PPN are drawn for the AMF only treatment. This is confusing, and also highlights that this data (PPN only) is missing in these figures.

We thank the Reviewer for this comment and have corrected the pictograms in Fig 4, whilst Fig 3 has been revised in line with other comments.

Addressed.

2. Transcriptional data in Figure 3: ...A more conventional way of presenting gene expression would be more appropriate and would allow the reader to interpret the data and judge the quality of the data. A heatmap for overview would be best, and plotting the normalized counts for important genes where differences between treatments are detected is required.

We have combined comments from Reviewer's 1 and 3 to change Fig 3 to aid clarity. We now present expression profiles for genes that are related to sugar transport that are differentially expressed in AMF-only roots compared to in AM-colonised roots with distal PPN infection. This indicates the distal impact of PPN on AM fungal-root interactions. We thank both reviewers for their constructive feedback. The main text figure is the recommended heatmap whilst the normalised counts for both sugar-related and fatty acid biosynthesis/transport-related genes (which are not significantly different between these two treatments) are provided in Fig S5, S6.

Adding the heatmap has improved the presentation of the data, as it is easier to interpret and compare genes between treatments. If you are basing your palmitic acid, glucose and fructose quantification on the fact that hexose-related but not fatty acid-related genes were differentially expressed, I would suggest to include all of these genes in one heatmap to see the whole picture. Please add the genes from Fig S6 (fatty acid-related genes) into the Fig. 3 heatmap.

A few other concerns/requests regarding the transcriptome data:

- Please add the PPN only samples to the Fig. S4 PCA plot for consistency with the rest of the RNA-seq data if you choose to include the data in Figure 3. **DONE**
- Please also convert Fig. S7 into a heatmap for consistency with the other data you have presented in the main figures. **DONE**
- RAM2, a mycorrhizal specific gene, is going up significantly in all of the treatments compared with the control which is interesting and consistent with the accumulation of palmitic acid in PPN only infected roots. Can you comment on why this may be? Has this been observed previously for PPN

infected roots? If presenting this data, it is worth mentioning and discussing this result in the manuscript. **DONE L358**

3. Palmitic acid measurements: While amounts of palmitic acid are high in plant roots, the amounts seem to be much lower when measuring fungal hyphae, and standard deviations appear very large.

The units used to express palmitic acid abundance in roots vs. hyphae are necessarily very different; per g root whilst for AM fungi it is expressed as per cm hyphae. We do not expect the amounts of any compound to be equally, or more, abundant in AM hyphae compared to root tissues given the direction of flow from plant to fungi. Contrary to the Reviewers comment, if the two values were indeed similar then we would expect an error in the experimental methods.

Apologies if my comment was not clear. I of course did not mean to suggest that you should find more lipids in fungal hyphae than in roots - I simply tried to emphasize that you are looking at very small amounts of palmitic acid and appear to have a very large variability between the data points.

With such small amounts and high variability between the data points, I question whether the authors are able to detect significant differences in palmitic acid transferred to AM fungi. The authors show that palmitic acid contents in PPN+AMF roots are reduced to control levels, however, in extraradical hyphae the same amounts of palmitic acids are present. This could be another sign for insufficient sensitivity in these measurements. However, one of the main conclusions of this paper is based on these measurements. This is the case for both nematode and aphid treatments. A more appropriate experiment would have been to trace carbon using ^{14}C labelled CO_2 and test how much of the label ends up in fungal hyphae, as either palmitic acid or sugars. The sensitivity would have likely been higher, and differences in the amounts of palmitic acid transferred would have been clearer.

LC-MS can detect very low quantities of palmitic acid, as shown by the standard curve provided in the figure below. The data here have been log transformed to demonstrate the ability of the method to detect, in this instance, 1ng of palmitic acid. In our experimental runs we extracted hyphae from approximately 200 cm of hyphae, detected the total compound abundance and then divided this by the hyphal lengths to generate ng/cm hyphae displayed in Figure 4 in our manuscript. Based on the data from Fig 4 in the manuscript, a minimum of approximately 15 ng of palmitic acid was detected in fungal hyphae, which is well over the detection limits of ~1ng suggested by the standard curve.

Standard curve of 1ng, 10ng, 100ng, 1000ng of palmitic acid detected via LC-MS.

The reviewer's suggestion to trace a $^{14}\text{CO}_2$ label into fungal hyphae and use this to identify compound identity would certainly be a novel approach, however it is technically intractable as the ^{14}C (half-life >5,000 years) would contaminate the instrument and be prohibitively expensive due to the replacement of instrument components after each run. Combined with the transcriptomic and isotope tracing data, the LC-MS data presented in our manuscript provides compelling evidence that there is a reduction of hexose sugar yet maintenance of palmitic acid flow to extraradical AM fungal hyphae when PPN co-colonise root systems.

Thank you for providing data on the detection limits of palmitic acid using LC-MS. I do not question that you are able to detect the low amounts of palmitic acids present in fungal hyphae. However, I still have concerns about the large degree of variance you see between replicates. There is a possibility that this variance is why no differences in palmitic acid are detected between the different conditions tested. One option to improve this might be to include more replicates for these measurements.

More generally however, measuring absolute amounts of metabolites only provides a snap shot of the metabolic state of a tissue. A possible interpretation based on the data you provide is that lipids accumulate in plant roots under different conditions, but that these lipids are not actually exported to AM fungi (this is under the assumption that you are not confidently detecting differences in palmitic acid levels between different treatments). The same is true for sugars. Based on the provided data (i.e. lower amounts of sugars in fungal hyphae), it is not possible to distinguish for example between a case where less sugars are exported to AM fungi, as opposed to a situation where the fungus might not break down the lipids it receives from plants into sugars because it uses lipids as longer-term carbon storage and for survival, particularly if overall much less carbon is being transferred to AM fungi, as your carbon tracing data (Figure 2) suggests. It is also possible that you observe lower amounts of sugars in fungal hyphae when PPN are present because the fungus metabolises these sugars more quickly. Overall, you are not directly

measuring flux/transport of metabolites, simply the amount present in different tissues.

The field has struggled to determine the relative amounts of carbon delivered to AM fungi in the form of lipids versus sugars for years. This is due to the technical difficulties associated with carbon labelling. One possible way around this would be to use carbon isotope tracing, as I suggested above. I understand that you do not want to use LC-MS to measure ¹⁴C labelled metabolites due to the long half-life of ¹⁴C. An alternative would be to use ¹³C. Importantly, one would have to do flux measurements, rather than just measuring absolute amounts of metabolites at one time point, to be able to measure export of metabolites from roots to AM fungi.

I would not want to suggest that you include these additional experiments for this publication, as these experiments are time consuming and difficult to do. However, if the current data is the only data you can provide, I strongly suggest that you include a detailed discussion on possible alternative interpretations of the metabolic data. In my opinion, you cannot conclusively show export of certain metabolites from plant roots to AM fungi based on the data you provide. This needs to be reflected in a more careful discussion of the findings.

We thank the Reviewer for these interesting insights and have included these in a discussion from L444.

4. In figure 3, SWEET1b is down-regulated in the PPN/AMF system compared to AMF only, and the authors suggest that this is the reason (along with downregulation of other SWEETs) for a reduction in sugar transfer to AMF. However, a reduction in sugar transfer is also apparent in PPN+AMF, based on sugar content in external hyphae, yet SWEET1b expression is not reduced. This inconsistency needs to be addressed.

We thank the Reviewer for asking for clarification on this. In joint PPN+AMF tissues we observed changes in gene expression derived from both parasitic and mutualistic interactions. Therefore, rather than an inconsistency, this indicates that as well as being potentially mycorrhizal-regulated, SWEET1b may additionally be regulated by the presence of parasitic nematodes. The extra data provided in Fig S5 suggests that this is true. We have added this interesting point to P17 L7.

Addressed.

5. The authors compare carbon allocation to AMF when either nematodes or aphids are present at the same time, and conclude from the data that a reduction in sugar allocation to AMF is a conserved mechanism. However, these are two very different pests, with nematodes infecting the roots, and aphids feeding on the shoots...

We agree with the reviewer. The stark contrasts between PPN and aphids was the rationale for choosing these pests to investigate conservation of the reduced sugar mechanism.

...This will have a major impact on fixed carbon and its allocation from shoots to roots. In the case of aphids, shoot carbon will be significantly reduced, meaning that less carbon is available to be allocated to AM fungi overall....

We thank the Reviewer for their comment, however this is not true. Shoot carbon is not significantly reduced during aphid feeding at ambient CO₂ levels (Charters et al., 2020, Current Biology, 30: 1801-1808).

Thank you for pointing out this reference. As far as I understand, Charters et al. 2020 investigated the effect of carbon allocation to AM fungi during aphid feeding in wheat. Is there evidence that shoot to root carbon allocation dynamics in your tested systems (*M. truncatula* and *S. tuberosum*) with aphids are the same as in wheat? Based on your changes on Page 18 lines 425-434, there do seem to be differences between species - am I interpreting this incorrectly?

We have amended this paragraph for clarity and thank you for pointing out this error. Charters et al 2020 did not analyse specific compounds so the similarity between Wheat and our study is currently unknown.

...In the case of nematodes, carbon allocation from shoots to roots could differ between AMF and nematode compartments. Unfortunately, no $^{14}\text{CO}_2$ experiments are shown for the aphid experiments, and so nothing can be concluded on the total amount of carbon that is allocated to roots with and without aphids.

Here, we show that carbon delivery to AMF and PPN compartments does differ (Figure 2a). However, regarding the reviewer's comment that "nothing can be concluded on the total amount of carbon allocated to roots with and without aphids"; again, this was not something we hypothesised or specifically tested in our experiments with aphids as previous studies have already shown total root carbon is greater in aphid infested plants than in non-infested plants (Charters et al., 2020). In this manuscript we investigated hexose and palmitic acid content of roots and AM external hyphae from +/- aphid treatments. Together, these results are very interesting considering we found similar amounts of palmitic acid yet reduced hexose quantities in roots from aphid infested plants compared to AM only plants and this may explain the observations regarding AM-carbon in Charters et al (2020). We have added this interesting discussion point to P18L2.

I agree that this is a very interesting observation, but would like to reiterate here that the authors need to be careful about conclusions that are drawn from the data provided regarding export of sugars and lipids to AM fungi, as discussed above. If the authors did not set out to test the hypothesis that less carbon is allocated to roots with and without aphids, this has to be made clear in the text (as suggested for comment 1).

Amended as per comment 1

5. Fungal colonization levels: Although authors quantify total colonization and state that colonization levels are the same across all treatments (Figure S2), it is unclear whether the infection structures, particularly arbuscules, remain the same and functional when parasites are present.

As part of the AM fungal colonisation assessment, we noted abundance of each AM structure and have amended Fig S2 to include this additional information. The functionality of this symbiosis is inferred by the response to the comment below, that there are comparable levels of plant shoot P between treatments including AM fungi, and our prior work (cited in the manuscript) has comprehensively quantified the transfer of labelled AM-acquired P into host plants (Bell et al., 2022, *New Phytologist* 234: 269-279). This is a far more accurate assessment of mycorrhizal function than observation and reporting of presence and abundance of fungal structures alone.

Addressed, now that the missing data for shoot total P in an AMF only infected plant has been provided.

Furthermore, In Figure S3, no data is provided on plant shoot total P in an AMF only infected plant, making it difficult to conclude whether as much phosphate is provided by AMF with additional parasites present.

We apologise for the oversight and have included these data for total shoot P in AM only plants in Fig S3 as requested. To conclude, we see no reductions in total shoot P when PPN are present.

Addressed.

1. It would be helpful to describe in the results section when (how many weeks after inoculation) colonization, metabolites, gene expression was measured.

We have copied this information from the Methods section to where relevant within the figure legends.

Addressed.

4. In the discussion, SWEET transporters are all described as transferring sugars to AMF. However, different SWEETs might have different functions. For example, some SWEETs might be involved in transferring sugars to arbuscule containing cells for lipid biosynthesis. It needs to be made clear that not all SWEETs will be directly involved in sugar transfer to fungal hyphae at the peri-arbuscular membrane, and that sugar allocation to roots might serve to provide substrates for lipid biosynthesis, not just for sugar provision to AM fungi.

We have added to P17L9 to include this interesting point.

Addressed.